# Innate Immune Pathways Promote Oligodendrocyte Progenitor Cell Recruitment to the Injury Site in Adult Zebrafish Brain

**DOI:** 10.3390/cells11030520

**Published:** 2022-02-02

**Authors:** Rosario Sanchez-Gonzalez, Christina Koupourtidou, Tjasa Lepko, Alessandro Zambusi, Klara Tereza Novoselc, Tamara Durovic, Sven Aschenbroich, Veronika Schwarz, Christopher T. Breunig, Hans Straka, Hagen B. Huttner, Martin Irmler, Johannes Beckers, Wolfgang Wurst, Andreas Zwergal, Tamas Schauer, Tobias Straub, Tim Czopka, Dietrich Trümbach, Magdalena Götz, Stefan H. Stricker, Jovica Ninkovic

**Affiliations:** 1Institute of Stem Cell Research, Helmholtz Center Munich, 85764 Oberschleißheim, Germany; rosario.sanchez@biologie.uni-muenchen.de (R.S.-G.); koupourtidou@helmholtz-muenchen.de (C.K.); lepko.tjasa@gmail.com (T.L.); alessandro.zambusi@helmholtz-muenchen.de (A.Z.); klara.novoselc@helmholtz-muenchen.de (K.T.N.); durovictamara90@gmail.com (T.D.); sven.aschenbroich@helmholtz-muenchen.de (S.A.); schwarz.veronika@campus.lmu.de (V.S.); magdalena.goetz@helmholtz-muenchen.de (M.G.); 2Department Biology II, University of Munich, 80539 München, Germany; straka@biologie.uni-muenchen.de; 3Biomedical Center (BMC), Division of Cell Biology and Anatomy, Faculty of Medicine, LMU Munich, 80539 München, Germany; 4Graduate School Systemic Neurosciences, LMU, 80539 Munich, Germany; 5Reprogramming and Regeneration, Biomedical Center (BMC), Physiological Genomics, Faculty of Medicine, LMU Munich, 80539 München, Germany; christopher.breunig@helmholtz-muenchen.de (C.T.B.); stefan.stricker@helmholtz-muenchen.de (S.H.S.); 6Epigenetic Engineering, Institute of Stem Cell Research, Helmholtz Center Munich, 85764 Oberschleißheim, Germany; 7Department of Neurology, Justus-Liebig-University Giessen, Klinikstrasse 33, 35392 Giessen, Germany; hagen.huttner@neuro.med.uni-giessen.de; 8Institute of Experimental Genetics, Helmholtz Center Munich, 85764 Oberschleißheim, Germany; martin.irmler@helmholtz-muenchen.de (M.I.); beckers@helmholtz-muenchen.de (J.B.); 9German Center for Diabetes Research (DZD e.V.), 85764 Neuherberg, Germany; 10Chair of Experimental Genetics, School of Life Sciences Weihenstephan, Technical University Munich, 80333 München, Germany; 11Institute of Developmental Genetics, Helmholtz Center Munich, 85764 Oberschleißheim, Germany; wurst@helmholtz-muenchen.de (W.W.); dietrich.truembach@helmholtz-muenchen.de (D.T.); 12Munich Cluster for Systems Neurology SYNERGY, LMU, 80539 Munich, Germany; 13Chair of Developmental Genetics c/o Helmholtz Zentrum München, School of Life Sciences Weihenstephan, Technical University Munich, 80333 München, Germany; 14German Center for Neurodegenerative Diseases (DZNE), Site Munich, 80539 Munich, Germany; 15Department of Neurology, Ludwig-Maximilians University, Campus Grosshadern, 81377 Munich, Germany; andreas.zwergal@med.uni-muenchen.de; 16Biomedical Center (BMC), Bioinformatic Core Facility, Faculty of Medicine, LMU Munich, 80539 München, Germany; tamas.schauer@med.uni-muenchen.de (T.S.); tstraub@bmc.med.lmu.de (T.S.); 17Centre for Clinical Brain Sciences, University of Edinburgh, Edinburgh EH8 9YL, UK; tim.czopka@tum.de; 18Biomedical Center (BMC), Division of Physiological Genomics, Faculty of Medicine, LMU Munich, 80539 München, Germany

**Keywords:** brain regeneration, oligodendrocyte progenitors, reactive gliosis, innate immunity pathways, zebrafish, neurogenesis, brain injury

## Abstract

The oligodendrocyte progenitors (OPCs) are at the front of the glial reaction to the traumatic brain injury. However, regulatory pathways steering the OPC reaction as well as the role of reactive OPCs remain largely unknown. Here, we compared a long-lasting, exacerbated reaction of OPCs to the adult zebrafish brain injury with a timely restricted OPC activation to identify the specific molecular mechanisms regulating OPC reactivity and their contribution to regeneration. We demonstrated that the influx of the cerebrospinal fluid into the brain parenchyma after injury simultaneously activates the toll-like receptor 2 (Tlr2) and the chemokine receptor 3 (Cxcr3) innate immunity pathways, leading to increased OPC proliferation and thereby exacerbated glial reactivity. These pathways were critical for long-lasting OPC accumulation even after the ablation of microglia and infiltrating monocytes. Importantly, interference with the Tlr1/2 and Cxcr3 pathways after injury alleviated reactive gliosis, increased new neuron recruitment, and improved tissue restoration.

## 1. Introduction

Wound healing in the brain is triggered by a temporarily regulated neuroinflammatory response that activates glial cells (reactive gliosis) and induces their recruitment to the injury [1,2]. Despite the many different approaches to model brain injury, there is an emergence of a common pattern in the cellular dynamics of brain resident cells following an insult [3]. Microglia respond to injury within 24 h by changing their morphology, increasing their proliferation rate, and migrating towards the injury site [4]. The activated microglia together with infiltrating monocytes not only phagocyte the cellular debris, but also release several damage-associated molecules (DAMs) to coordinate the subsequential glial reactivity [3,5,6,7,8]. In line with the inductive role of microglia released DAMs, the astrocyte reactivity (also termed astrogliosis) typically starts later on (2–3 days after injury) and varies depending on the extent of the damage [9]. Astrogliosis forms the border of GFAP^+^ reactive astrocytes surrounding the injury site by the hypertrophy of astrocytic processes, the upregulation of GFAP, and the increased proliferation of juxtavascular astrocytes [5,10,11]. Ablation experiments have demonstrated that the initial reaction of astrocytes is essential for wound closure and the restoration of the normal brain milieu [12,13]. Moreover, border-forming reactive astrocytes have been described to be necessary for axonal regeneration after spinal cord injury [5,14]. On the other hand, multiple studies have shown that prolonged astroglial reactivity induced aversive extracellular matrix modifications and exacerbated inflammation [15,16,17] that negatively impact on functional recovery. Recent reports have demonstrated a reciprocal regulatory loop between astrocytes and immune cells. While DAMs released by microglia induce a neurotoxic phenotype in astrocytes [18], astrocytes regulate the extravasation of monocytes and in turn, the long lasting neuroinflammatory response [12,19].

Strikingly, oligodendrocyte progenitor cells (OPCs) react to insults such as demyelination [20,21], traumatic brain injury (TBI) [22], or neurodegenerative disorders [23] as fast as the residential microglial cells. In physiological conditions these slow proliferating progenitors display very limited and short-range migration [24,25] and maintain their non-overlapping cellular domains by balancing cellular proliferation and cell death [22,24,25,26,27]. However, a rapid and heterogeneous reaction of OPCs has been documented in response to brain injury [22,24,25,28]. The OPCs polarize shortly after an insult [29] and become fully hypertrophic [24,25] within 48 h after injury. This reaction is followed by migration towards the injury site [24,25] and an increased proliferation rate in the case of a bigger injury [25]. Importantly, during the wound healing process the OPCs do not maintain the non-overlapping domains anymore and they accumulate at the injury site [25]. This accumulation is resolved 4 weeks after the injury and the cellular repulsion mechanisms maintaining the non-overlapping domains are established again. Despite the precise description of the cellular dynamics, our understanding of the OPC reaction to brain injuries and its relevance is still far from being understood. Several reports have suggested that the OPC accumulation at the injury site could promote wound healing [25] and the co-depletion of proliferating microglia and OPCs shown enhanced axonal regeneration [30,31]. In contrast, the accumulation of OPC-derived proteoglycan NG-2 has been associated with the inhibition of axonal growth [32]. These rather opposing observations support the need to identify the pathways regulating the reaction of OPCs to brain injury and to associate them with the observed regenerative outcomes after injury.

The temporal sequence of the glial reaction to injury in the adult zebrafish brain shares some interesting similarities with the injured mammalian brain [1,33,34]. However, in contrast to the exacerbated gliotic wound closure described above for the mammalian brain, the glial response leads to full tissue restoration in zebrafish [1,34,35,36,37]. Complete recovery in the zebrafish brain correlates with the capacity to regulate the neuroinflammatory landscape and induce the restorative neurogenesis (neuronal replacement) from endogenous sources [37]. The basis for neuronal replacement resides in the injury-induced activation of neural stem cell-like ependymoglial cells [1]. The initial microglial reaction to injury activates developmental and/or injury-specific regulatory pathways in ependymoglial cells [1], regulating the timely production of new neurons necessary for tissue recovery. Although several regulatory mechanisms mediating the crosstalk between immune and ependymoglial cells have been identified [1], little is known about how OPCs contribute, if at all, to the permissive time window for the integration of new neurons. Zebrafish OPCs exhibit a different reaction to the stab wound injury depending on the injury paradigm [37,38,39]. While injuries performed rostro-caudally through the nostrils (nostril injury) induced no increased proliferation and no recruitment of OPCs to the damaged area [37], injuries along the dorso–ventral axis (skull injury) induced a long lasting accumulation of OPCs at the injury site [39]. The reaction of the OPCs to the skull injury is indeed very similar to the OPC reaction in the injured mammalian cortex, including the temporal resolution only after 4 weeks [22,34,39]. Nostril and skull injury paradigms offer an ideal comparative model by which to identify specific molecular pathways regulating OPC reactivity and its potential impact on tissue restoration and neuronal replacement as a means for functional recovery. We applied a comparative analysis of “nostril versus skull” zebrafish forebrain tissue to identify novel molecular mechanisms regulating exacerbated and prolonged OPC activation. We identified the toll-like receptor 2 (Tlr2) and the chemokine receptor 3 (Cxcr3) innate immune pathways as key regulators of OPC proliferation. Interference with these signaling pathways after injury not only alleviated the OPC accumulation at the injury site, but it also improved wound healing and restorative neurogenesis. We also showed that prolonged exposure of murine OPCs to human cerebrospinal fluid content activated Tlr2/Cxcr3 signaling and in turn increased OPC proliferation. Taken together, we identified signaling pathways and the source of their ligands regulating exacerbated and prolonged OPC reactivity, opening a new avenue for targeting therapies.

## 2. Materials and Methods

### 2.1. Animals

Adult 4–6 month old wild-type zebrafish (Danio rerio) of the AB/EK strain, or of the transgenic lines, Tg(olig2:gfp) [40], Tg(olig2:DsRed) [40], Tg(fli1:egfp) [41], Tg(gfap:GFP) [42], Tg(mbp:nslGFP) [43] and Tg(mpeg1:mCherry) [44] were used for all the experiments. Fish were kept under standard husbandry conditions [45] and experiments were performed according to the handling guidelines and regulation of the EU and the Government of Upper Bavaria (AZ 55.2-1-54-2532-09-16).

### 2.2. Stab Wound Injuries

We carried out three different stab wound injury paradigms: nostril, skull, and small skull injuries (Figures 1 and 6). Fish were anesthetized with buffered 0.02% MS-222 for 45 s to a minute and then placed in a Tricaine-soaked sponge. With the visual aid of a dissecting microscope, injuries were performed in both telencephalic hemispheres. The nostril injury [37] was performed using a 100 × 0.9 mm glass capillary needle (KG01, A. Hartenstein). Capillaries were pulled on a Narishige Puller (model PC-10) using a “One-stage” pull setting at a heater level of 63.5 °C. The final dimensions of the capillaries were 5 mm in length and 0.1 mm in diameter. For the skull injury, a micro-knife (Fine Science Tools) was inserted vertically through the skull into the medial region of the telencephalon. To perform the small skull injury, the skull was thinned above the telencephalon area using a micro-driller (Foredom) and the glass capillary (identical to that used for the nostril injury) was inserted vertically through the skull and brain parenchyma. After the injury, fish were placed in fish water with oxygenation to assure complete recovery from the anesthesia.

### 2.3. Tlr2 Agonist (Zymosan A Bioparticles, Invitrogen) Administration

Fish were anaesthetized in 0.02% MS-222 and a small hole, using a micro-knife (Fine Science Tools), was made into the skull (above the telencephalic ventricle). A glass capillary loaded with 10 mg/mL Zymosan or artificial cerebrospinal fluid with Fast Green dye to visualize the injection site (0.3 mg/mL; Sigma) was inserted into the hole and ~1 μL of solution was injected at a pressure of 150 hPa using a microinjector (Eppendorf, Hamburg, Germany). Artificial cerebrospinal fluid was used as a control for the ventricular injections since its composition closely matches the electrolyte concentrations of cerebrospinal fluid (Figure 2).

### 2.4. Cxcr3 Agonist (VUF 11222, R & D Systems) Administration

Fish were anaesthetized as previously described and the solution was injected intraperitoneally using a 30 G needle (Braun). A total of 4 μL of VUF 11222 (300 mg/kg) or DMSO vehicle with the Fast Green dye (0.3 mg/mL; Sigma) was intraperitoneally injected (Figure 2). We did not observe any aversive effects by IP injecting up to 4 μL of the 80% DMSO solution.

### 2.5. Inhibitor Administration

Intraperitoneal injections were performed as described above. All inhibitors were dissolved in DMSO with Fast Green dye (0.3 mg/mL; Sigma) for visualization. Cxcr3 inhibitors (NBI 74330 (300 mg/kg, R & D Systems, Minneapolis, MN, USA) and AMG 487 (112 mg/kg, R & D Systems)), and Tlr1/2 inhibitor (CU CPT 22 (150 mg/kg, R & D Systems)) were injected independently (NBI 74330 or CU CPT 22) or in combination (NBI 74330 + CU CPT 22 or AMG 487 + CU CPT 22) (Figure 3 and Appendix A). NBI 74330 and CU CPT 22 were injected daily, while AMG 487 was injected twice per day. The vehicle solution consisted of DMSO and Fast Green dye. A total of 4 μL was administrated per injection and the maximum number of intraperitoneal injections was 2 injections per day at an interval of 72 h. The mortality rate was less than 5% after any of the treatments.

### 2.6. BrdU Labelling Experiments

To label proliferating cells and their progeny, we carried out long term bromo-deoxy-uridine (BrdU; Sigma Aldrich, St. Louis, MO, USA) incorporation. Fish were kept in BrdU-containing aerated water (10 mM) for 21 h/day or 14 h/day as stated in specific experiments (Figures 4–6).

### 2.7. Immune Cell Depletion Assay

A two-step approach was used to deplete the immune cells: 2 μL of Clodrosome (Encapsula NanoSciences, Brentwood, TN, USA) was injected into the telencephalic ventricle every second day for one week prior to the injury (4 injections in total prior to the injury). Ccr2 inhibitor (MK-0812, 82.5 mg/kg, Cayman Chemical, Ann Arbor, MI, USA) was administered by intraperitoneal injection daily, starting 2 days before the injury (Figure 5 and Appendix A). As a control for the ventricular injection, we used empty liposomes (Encapsome) and DMSO for the intraperitoneal administration.

### 2.8. Human CSF Sample Collection

Human CSF samples were obtained from two different sources. The first one was Erlangen University Hospital. The patients underwent a lumbar puncture to exclude intracranial hemorrhage or inflammatory diseases of the CNS and they were considered healthy based on normal values for CSF (color, cell count, and total protein). CSF analysis was approved by the institutional review board of Erlangen University Hospital (ethics committee number 3950) and patients gave informed consent. After lumbar puncture, a protease inhibitor was added to CSF according to the manufactureR′s instructions (Roche) and CSF was directly frozen at −80 °C. The second source was the University Hospital at LMU Munich (project number 159/03). Human CSF samples were obtained from patients who underwent a lumbar puncture to exclude intracranial hypertension or inflammatory CNS diseases. Routine analysis of CSF (cell count, total protein, glucose) revealed no abnormal values in all samples. All patients gave their informed written consent.

### 2.9. Human Plasma, Cerebrospinal Fluid, and Heat-Inactivated Cerebrospinal Fluid Administration

A 100 × 0.9 mm glass capillary needle (KG01, A. Hartenstein, Würzburg, Germany) was loaded with human plasma (Sigma Aldrich), human cerebrospinal fluid, or heat-inactivated human cerebrospinal fluid (single healthy donor). Human cerebrospinal fluid was incubated for 15 min at 90° to generate heat-inactivated human cerebrospinal fluid. Fish were anesthetized with 0.02% MS-222 (Sigma-Aldrich). The glass capillary was introduced through the nostril and ~1 μL of the solution was injected at a pressure of 150 hPa into the injury track in the telencephalic parenchyma (Figure 6).

### 2.10. Plasmid Electroporation

The plasmid pCS2-tdTomatomem was diluted in sterile water and Fast green (1 mg/mL), reaching a final concentration of 1 μg/μL. ~0.5 μL of the solution was injected in the telencephalic ventricle as described previously [46]. Next, the electroporation was carried out by placing the positive electrode at the ventral side of the fisH′s head and the negative electrode on the dorsal side and giving five pulses at 40 V for 50 ms each at 1-s intervals [47].

### 2.11. Tissue Preparation and Immunohistochemistry

Animals were sacrificed by an MS-222 overdose. Brains were dissected and fixed for 3 h at 4 °C in 4% paraformaldehyde (PFA) in phosphate-buffered saline (PBS), washed three times with PBS, and sectioned. For sectioning, whole brains were embedded in 3% agarose in PBS and cut serially at a 100 µm thickness with a microtome (HM 650 V, Microm). Primary antibodies (Appendix A) were dissolved in 0.5% Triton X and 10% normal goat serum. Subclass-specific secondary antibodies (1:1000, Thermofisher, Waltham, MA, USA) were used to detect the primary antibodies. Nuclear staining was performed with 40,6-diamidino-2-phenylindole dihydrochloride (DAPI) (Sigma). All sections were mounted using Aqua Polymount (Polyscience, Niles, IL, USA). BrdU immunodetection required 2N HCl pre-treatment for 20 min at room temperature. Pre-treatment of the sections with Dako target retrieval solution (Agilent, Santa Clara, CA, USA) was necessary for the detection of the L-plastin. For whole-mount infarct tissue imaging, 500 µm thick telencephalic sections were cleared using BABB (1 part benzyl alcohol, 2 parts benzyl benzoate method) and stained as previously described [48]. 

Cryo-sectioning was used for RNAscope (see below). After fixation, whole brains were cryoprotected in a 30% sucrose solution overnight at 4 °C. The tissue was embedded in a tissue freezing medium (Leica) and frozen using dry ice. Serial sectioning at 20 µm thickness with a cryostat (Leica) was performed. Sections were stored at −20 °C until further processing.

### 2.12. RNAscope

We used an RNAscope Multiplex Fluorescent Reagent Kit v2 (ACD) to identify and label specific zebrafish RNAs. For tissue processing, pre-treatment, and RNAscope assay we followed the manufactureR′s instructions. The RNAscope probes were designed by ACD using the following target sequences: Cxcr3.2 (NM_001007314.1), MYD88 (NM_212814.2), Mxc (NM_001007284.2), Tlr8b (NM_001386709.1), and GFP (Synthetic construct Cox8ND6gfp). For visualization, the TSA Plus Cyanine 3/5 (Perkin Elmer) kit was utilized.

### 2.13. Image Acquisition and Processing

All immunofluorescence microscopy on sections was performed and analyzed with an Olympus FV1000 cLSM system (Olympus, Tokyo, Japan), using the FW10-ASW 4.0 software (Olympus). Bright field images were taken with a Leica DM2500 microscope at the Core Facility Bioimaging at the Biomedical Center (BMC). For whole-mount infarct tissue analysis, images were acquired with a Leica SP8X microscope, using LASX software (Leica) and deconvolved using Huygens Professional deconvolution software (SVI). The injury site was analyzed using Imaris software version 8.4 (Bitplane, Concord, USA). The 3D surface object was generated from manually created contours on 2D slices using the Surface tool to calculate the volume of the Surface object. Animations were made using the Key Frame Animation function (Imaris).

### 2.14. Quantitative Analysis

For each experiment, animals were randomly distributed into groups and all manual counts were performed blind. For all quantifications 4 to 6 brains were analyzed, coming from at least 2 independent experiments. All the sections belonging to the telencephalon were quantified (sections with the olfactory bulb or optic tectum were excluded), from which we analyzed the entire rostro-caudal extent of the injured tissue. The injured volume was calculated by multiplying the area and the depth of the DAPI dense accumulation for each section. The total injured volume was the sum of all the injured sections. The density of the positive cells in the injured volume was defined by the total number of cells located in the volume occupied by DAPI dense accumulation. Controls for the “injured volume” were measured in uninjured, age matched fish using equivalent volumes for each of the injury paradigms. For 4C4 quantifications. single-channel immunofluorescent images were converted to black and white, thresholded, and the extent of the stained area was measured using NIH ImageJ software. For the analysis of OliNeu proliferation, 25 randomly selected images per coverslip were used for the analysis. The analysis was performed using the automated ImageJ pipeline that is available upon request.

### 2.15. Statistical Analysis

Data are presented as the mean +/− standard error of the mean (SEM) and each dot represents one animal. Statistical analysis was performed using R (version 3.6.1). Data were investigated to test whether assumptions of parametric tests were satisfied (e.g., *t*-test or Anova). Residuals (fitted by lm function, stats package, version 3.6.1) were tested for normality by the Shapiro–Wilk normality test (shapiro_test function, rstatix package, version 0.6.0). Further diagnostics of residuals were carried out using the DHaRMa package (version 0.3.3.0). The homogeneity of variance assumption was tested using Levene’s test (leveneTest function, car package, version 3.0–10). If both normality and equal group variances assumption were met, Student’s *t*-test (*t*.test function with var.equal = TRUE, stats package, version 3.6.1) for single comparisons and one-way anova (aov function, stats package, version 3.6.1) for multiple group design was used. Anova post-hoc tests, i.e., Tukey or Dunett tests, were applied either for all pair-wise comparisons (tukeyTest function, PMCMRplus package, version 1.9.0) or Many-to-One comparisons (dunnettTest function, PMCMRplus package, version 1.9.0), respectively. If the normality assumption was satisfied but groups had unequal variances, WelcH′s *t*-test (*t*.test function with var.equal = FALSE, stats package, version 3.6.1) for single comparisons and WelcH′s one-way Anova (oneway.test function, stats package, version 3.6.1) for multiple group design was used. As a post-hoc test, Dunnett’s T3 test for data with unequal variances was applied (dunnettT3Test function, PMCMRplus package, version 1.9.0). For Appendix A only selected contrasts were tested (i.e., coronal vs. sagittal in each group) using the multcomp package (glht function, version 1.4–15). If the normality assumption was not met, the data were log-transformed to achieve normality of the residuals. In such a case, parametric tests were carried out as described above. If log-transformation did not satisfy the assumption, non-parametric tests were used i.e., Wilcoxon rank sum test (wilcox.testfunction, stats package, version 3.6.1) for single comparisons and Kruskal–Wallis test (kruskal_test function, rstatix package, version 0.6.0) for multiple group design. In the latter case, the post-hoc Dunn test (kwManyOneDunnTest function, PMCMRplus package, version 1.9.0) was performed for Many-to-One comparisons. Dose-response in Figure 7D was analyzed by linear regression on square-root transformed outcome values. *p*-values were obtained for the regression coefficients: the slope for OliNeu and the difference in slopes (interaction term) for clone1 or clone2 relative to OliNeu. The detailed statistical analysis for all data sets is presented in the Appendix A.

### 2.16. Analysis of Restorative Neurogenesis

Restorative neurogenesis was defined as the proportion of new neurons that migrated into the parenchyma. Zebrafish were kept in BrdU-containing aerated water (10 mM) overnight during the first 3 days. Simultaneously, vehicle and double inhibitors were injected daily under normal conditions (Figure 4C–H) or after immune cell depletion (Figure 4M–R). Animals were sacrificed at 7 dpi and the expression pattern of HuC/D and BrdU was analyzed. We assessed restorative neurogenesis as completed previously [49] by calculating the proportion of new neurons (HuC/D^+^BrdU^+^) that had migrated from the ventricular zone into the parenchyma as a result of the injury.

### 2.17. RNA Extraction, cDNA Synthesis, and RT-qPCR

Total RNA was isolated using the Qiagen RNeasy kit for microarray analysis and qPCR experiments. RNA isolation from FACS purified was performed with a PicoPure RNA isolation kit (Thermo Scientific). cDNA synthesis was performed using random primers with the Maxima first strand synthesis kit (Thermo Scientific). The manufactureR′s instructions were followed for all the mentioned kits. The real-time qPCR was conducted using SYBR green and Thermo Fisher Quant Studio 6 machine (Appendix A). 

### 2.18. Microarray Analysis

Total RNA (20 ng) was amplified using the Ovation Pico WTA System V2 in combination with the Encore Biotin Module (Nugen). Amplified cDNA was hybridized on Affymetrix Zebrafish 1.0 ST arrays. Staining and scanning were performed according to the Affymetrix expression protocol including minor modifications as suggested in the Encore Biotion protocol. An expression console (v.1.3.1.187, Affymetrix) was used for quality control and to obtain annotated normalized RMA gene-level data (standard settings including median polish and sketch-quantile normalization). Statistical analyses were performed by utilizing the statistical programming environment R (R Development Core Team implemented in CARMAweb [50]). Genewise testing for differential expression was performed by employing the limma *t*-test. Regulated gene sets were defined by *p* < 0.05, fold-change > 1.6x and linear average expression in at least one group >20. The array data have been submitted to the GEO database at NCBI (GSE98217).

### 2.19. Assignment of Zebrafish Array Probes to Homologous Mouse Genes

The genomic positions of all probe sets in the presented zebrafish microarray study were extracted from Affymetrix (http://www.affymetrix.com/analysis/index.affx; accessed on 24 November 2016) by applying a Batch Query on the GeneChip Array “Zebrafish Gene 1.x ST” (genome version Zv9 from 2011). With the help of a custom-written Perl script and the extracted genomic positions of the probe sets, zebrafish gene identifiers were derived from the Ensembl database via the Application Programming Interface (API), version 64, and subsequently passed to the Ensembl Compara database in order to retrieve homologous mouse genes. The Compara database stores pre-calculated comparative genomics data of different species including information on homologous genes, protein family clustering, and whole genome alignments [51]. For the assignment of zebrafish to mouse genes, all kinds of homology (i.e., one-to-one, one-to-many and many-to-many orthologous genes) were taken into account. Gene Ontology enrichment analyses were performed using the equivalent mouse symbols and DAVID Bioinformatics Resources 6.7 (*p*-value 0.05, fold change > 2) [52,53].

### 2.20. FACS Analysis

Animals from the Tg(olig2:DsRed) transgenic lines were sacrificed by an MS-222 overdose and the telencephalon was dissected from each animal. A single cell suspension was prepared according to previously published protocols [54,55] and cells were analyzed using a FACS Aria III (BD) in BD FACS Flow TM medium. Debris and aggregated cells were gated out by forward scatter–sideward scatter; single cells were gated in by FSC-W/FSC-A. Gating for fluorophores was performed using AB/EK animals. Cells were directly sorted into an extraction buffer from PicoPure RNA isolation kit (Thermofisher) and stored at −80 °C until RNA preparation was performed.

### 2.21. Preparation of Libraries for Deep Sequencing

cDNA was synthesized from 1 ng of total RNA using SMART-Seq v4 Ultra Low Input RNA kit for Sequencing (Clontech), according to the manufactureR′s instructions. The quality and concentration of cDNA was assessed on an Agilent 2100 Bioanalyzer before proceeding to the library preparation using a MicroPlex Library Preparation kit v2 (Diagenode). All libraries (minimum of 3 biological replicates per condition) were processed together to minimize batch effects. Final libraries were evaluated and quantified using an Agilent 2100 Bioanalyzer and the concentration was measured additionally with a Quant-iT PicoGreen dsDNA Assay Kit (Thermo Fisher) before sequencing. The uniquely barcoded libraries were multiplexed onto one lane, and 150-bp paired-end deep sequencing was carried out on HiSeq 4000 (Illumina) that generated approximately 20 million reads per sample.

### 2.22. RNAseq Analysis

The RNA-seq analysis was completed using the kallisto pipeline for the reads mapping and quantification followed with the Sleuth pipeline for the statistical analysis. The cut-off for the differentially regulated genes was based on the expression fold change (>2 fold) and *p*-value adjusted for the 10% false discovery rate (*q*-value < 0.05). FastQ files are deposited at (accession number pending). Gene Ontology enrichment analyses was performed using DAVID Bioinformatics Resources 6.8 (*p*-value 0.05, fold change > 2) [52,53].

### 2.23. Primary OPC Culture and Clonal Analysis

Primary cultures of the OPCs were performed as previously published [56]. Cortices of P0 mice were dissected avoiding the inclusion of white matter and grown for 10 days. After the initial culturing, cells were plated in 24-well plates at 727 cells/mm^2^ density. OPC primary cultures were transduced with a GFP encoding MLV-based virus for clonal analysis as previously described [57]. A total of 12 hrs after the transduction, the cells were treated with 1 µM NBI 74330 and 8 µM CU CPT 22 and analyzed 5 days later and the clonal analysis was performed as described previously [57].

### 2.24. Generation of gRNAs for CRISPR/Cas9-Mediated Deletion

Target sequences were chosen within 600 bp after the first ATG of the ORF of Cxcr3 (ENSMUSG00000050232) and Tlr2 (ENSMUSG00000027995). gRNAs were generated using Benchling (www.benchling.com, accessed on 24 September 2019) and chosen according to a high (>30) specificity score. Multiplexed gRNA vectors were generated using the STAgR protocol [58]. Single gRNA expression units were amplified using overhang primers, employing the N20 targeting sequence as homology for Gibson assembly. gRNAs were assembled into a gRNA expression vector containing a TdTomato reporter, modified after pgRNA1 [59].

### 2.25. DNA Extraction and PCR

For the DNA extraction from the cells, the DNeasy blood and tissue kit was used (Qiagen, 69504). The region containing the prospective mutation was amplified using the standard PCR condition (denaturation: 20″; annealing 20″; extension 60″; 30 PCR cycles) and locus-specific primer pairs from the positive and negative clones. Tlr2: 5′-ggacaaattcaggaagcgca and 5′-tgagagatcacggaccaagg; Cxcr3: 5′-cctcatagctcgaaaaacgcc and 5′-ccccggagagaaagagtcag. PCR products were cloned using a PCR cloning kit according to the manufactureR′s instructions (Stratagene) and were analyzed for the mutation using SANGER sequencing.

### 2.26. Generation of the Oli-Neu Cell Line Deficient for Cxcr3 and Tlr2

Oli-Neu cells were cultured in a SATO medium containing 1% horse serum. For each transfection, 200,000 cells/well were seeded into 6-well plates and coated with poly-L-lysin (Sigma). A total of 1 μg of each STAgR (encoding for gRNAs and TdTomato reporter) or control plasmid (encoding for dsRed) in addition to 1 µg of Cas9 plasmid (with a puromycin resistance cassette) was transfected per sample using Lipofecatmin 2000 (Invitrogen, Waltham, MA, USA) according to the manufactureR′s instructions. Cells were plated in low density and selected with 0.8 µg/mL puromycin for Cas9 expression. Five days later, clones transfected with STAgR (TdTomato reporter^+^, positive clones) or only Cas9 (TdTomato reporter^−^, negative clones) were selected and expanded. The proliferation analysis was performed using two independent clones with different deletions in both Tlr2 and Cxcr3 genes. To analyze the clones, 25,000 cells/well were plated in 24-well plates on poly-L-lysine-coated coverslips and analyzed after 48 h. Cell were fixed in 4% paraformaldehyde (PFA) in phosphate-buffered saline (PBS) for 15 min at room temperature and processed for the antibody staining.

### 2.27. Screen for Cxcr3 Ligands from the CSF

As the screen requires many cells, we decided to conduct it in the oligodendrocyte progenitor line (OliNeu) that also allows for the genetic inactivation of Cxcr3 and Tlr2 as described in Section 2.26. Both WT Oli-Neu and Cxcr3 and Tlr2 deficient clones were expanded onto a SATO medium containing 1% horse serum. After expansion, cells were re-plated on PLL-coated coverslips at an equal density (272 cells/mm^2^) and cultured for 2 h. After this pre-incubation, cells were treated with different CSF concentrations and vehicles. All cytokine treatments were performed using the WT Oli-Neu cells in quadruplets and at 3 different concentrations (Appendix A) that were used as independent replicates for the analysis. Cells were fixed with 4% PFA 24 h after the treatment and assessed for proliferation using the anti-PH3 immunostaining.

### 2.28. Human Cytokine Antibody Array

Four cerebrospinal fluid samples, derived from healthy patients, were analyzed using a Human Cytokine Antibody Array (abcam, ab133997). All samples induced a scarring reaction upon injection into the nostril injury track. The positive controls were used to normalize signal responses across multiple arrays.

## 3. Results

### 3.1. Skull and Nostril Models of Zebrafish Telencephalon Injury Differ in the Kinetics of the Glial Reaction

To identify the molecular and cellular basis for OPC activation during wound healing, we set out to follow the reaction of different cell types to an injury in the zebrafish telencephalon using two paradigms in parallel, one with long-term, exacerbated OPC reactivity (referred to as skull injury, Figure 1A) and the other resulting in time-restricted gliosis and full tissue recovery (referred to as nostril injury, Figure 1B). Injuries were performed in both telencephalic hemispheres and the injury site was defined based on the DAPI accumulation throughout the manuscript (e.g., Appendix A). Damage-associated molecules trigger the early inflammatory response that induces the recruitment of peripheral neutrophils into the injury site in the mammalian brain [60]. In zebrafish, we observed Lys^+^ neutrophils 12 h after both injuries in the brain parenchyma (Appendix A). Interestingly, Lys^+^ cells accumulated at the injury site after the nostril injury (Appendix A), while they were dispersed throughout the injured parenchyma after the skull injury (Appendix A). Moreover, Lys^+^ neutrophil accumulation resolved 24 h after nostril injury and we could not detect any difference 48 h after injury (Appendix A) compared to the intact brain. In contrast, we did not observe the fast clearance of Lys^+^ cells after skull injury (Appendix A).

As neutrophils regulate the activity state of microglia and extravasating monocytes and consequently the regenerative response [60], we analyzed both populations based on the expression of two different immunohistochemical markers (4C4 and L-plastin) as well as the transgenic line Tg(mpeg1:mCherry) [44] labelling both cell types (Appendix A). In the intact condition, the majority of microglia co-expressed all three markers although at the different levels (Appendix A). However, after both nostril and skull injury, we observed an increase in 4C4^+^ cells and only a proportion of them were colocalized with L-plastin and/or with mpeg1:mCherry^+^ cells (Appendix A). Taken together, these data suggest that 4C4 was the broadest marker to identify microglia/monocyte population and, therefore, we used it further in our study. While the initial activation pattern of 4C4^+^ cells was similar in both injury paradigms with the first signs of reactivity detectable already at 24 h after injury (Appendix A), the skull injury induced a stronger reactivity and a long-lasting accumulation of 4C4^+^ cells at the injury site (Figure 1E–M′).

The accumulation of cells belonging to the oligodendrocyte lineage (OPCs and mature oligodendrocytes were labeled using the transgenic line [Tg(Olig2:GFP)] [40]) at the injury site was slightly delayed in comparison with the microglia/monocytes (Appendix A). The density of Olig2:GFP^+^ cells was increased at the injury site 3 days after both skull (Figure 1F,G,N) and nostril injury (Figure 1H,I,O), although to different extents. The accumulation of Olig2:GFP^+^ cells was rapidly resolved and returned to pre-injury conditions within 7 days after the nostril injury (Figure 1L–M′,O), in agreement with previously published studies [33,37,38]. In contrast, the density of Olig2:GFP^+^ cells further increased and still persisted at 7 days post-injury (dpi) in the skull injury paradigm (Figure 1J–K′,N). We analyzed coronal brain sections depicting the skull injury in its full extent, but only part of the nostril injury. Therefore, the accumulation of both 4C4^+^ and Olig2:GFP^+^ cells observed exclusively after skull injury could be a consequence of a bias in the analysis. To exclude any technical bias, the number of Olig2:GFP^+^ cells accumulating at the nostril injury site was also analyzed in sagittal sections depicting the full extent of the nostril injury (Appendix A). No differences were observed at any of the analyzed time points (Appendix A). Moreover, injury sites were analyzed in BABB-cleared brains. While we could observe a clear accumulation of Sox10^+^ (classical marker for the oligodendrocyte lineage) and 4C4^+^ cells 3 days after nostril injury, 7 days after injury Sox10^+^ and 4C4^+^ cells showed distributions that were indistinguishable from samples of intact brains (Appendix A). 

In the zebrafish telencephalon the resident neural stem cell [48,61], the ependymoglial cells, express GFAP. So next, we used the Tg(gfap:GFP) transgenic line [42] to label and characterize the reactivity of ependymoglial cells after both types of injury. Gfap:GFP^+^ cell bodies line up at the ventricular wall of the brain surface with processes reaching basement membrane (Appendix A); therefore, after a nostril injury, only some processes of ependymoglia, located in the deep parenchyma, were wounded (Appendix A). Importantly, no sign of damage was observed at 7 days after nostril injury (Appendix A). On the other hand, upon skull injury, the ependymoglial cell layer was disrupted (Appendix A), but was already restored 7 days after skull injury (Appendix A). Despite this recovery, we still observed hypertrophic processes and a few misplaced Gfap:GFP^+^ cells at the injury site (Appendix A). 

The accumulation of Olig2:GFP^+^ and immune 4C4^+^ cells was resolved 28 days after the skull injury (Figure 1P), resembling the behavior of OPCs and microglia in the injured mammalian cerebral cortex [19,22]. However, even after the accumulation of Olig2:GFP^+^ and immune 4C4^+^ cells was resolved, the tissue architecture was not fully restored, based on the Gfap:GFP^+^ ependymoglial cell morphology (Figure 1Q–R′). To assess the ependymoglial morphology, we labelled them using the electroporation of the TdTomatomem plasmid both after nostril and skull injury and analyzed their morphology and localization 28 dpi. In line with previous reports [37,47], the nostril injury did not change the morphology or the localization of ependymoglial cells compared to the intact brain. We found ependymoglial cell bodies lining up at the telencephalic ventricular wall with processes mostly spanning the brain parenchyma and anchoring at the basement membrane 28 dpi (Figure 1R,R′). However, after skull injury, several of the labelled ependymoglial cells had a bushy morphology and did not reach the basement membrane (Figure 1Q; Appendix A).

These data demonstrate the differential reactions of neutrophils, microglia/monocytes, and oligodendrocyte lineage cells in two injury paradigms. The prolonged reaction of these cells correlates with the deley in the tissue restoration.

### 3.2. Activation of Innate Immunity Pathways Induces Prolonged Glia Reactivity after Injury in the Zebrafish Telencephalon

In view of the above findings, comparing the transcriptome induced by nostril and skull injury offers a unique opportunity to disentangle the specific molecular programs inducing exacerbated gliosis from the beneficial pathways promoting wound healing. We reasoned that some signaling pathways that were activated after a skull injury, but not after a nostril injury, could account for the long-lasting glial accumulation at the injury site and the absence of full tissue restoration. Therefore, we analyzed the gene expression during regeneration (1, 2, 3, and 7 dpi) after a nostril or skull brain injury in the whole telencephalon, using the Affymetrix Zebrafish Gene ST 1.0 array (Figure 2A). Both types of injuries initially induced comparable transcriptome changes, as reflected by a similar number of significantly regulated genes (fold change > 1.6, *p* < 0.05) and a large overlap in significantly overrepresented Gene Ontology (GO) terms (based on DAVID analysis, fold enrichment ≥ 2; *p* < 0.01) at 1 and 2 dpi (Figure 2B, Appendix A). However, we observed a striking difference in the number of regulated GO terms after nostril and skull injury at 3 dpi (Figure 2B), with 1012 transcripts regulated after a skull but not nostril injury (Figure 2C). Interestingly, this large number of uniquely regulated genes at 3 dpi correlates with differences in the reaction of Olig2:GFP^+^ cells and microglia/monocytes between the two injury paradigms (Figure 1N,O), supporting the idea that understanding these transcriptional differences could identify specific programs inducing long-lasting OPC accumulation and neuroinflammation. To further validate the applicability of this approach, we analyzed the differential expression of genes possibly involved in the ECM modifications, as the specific ECM changes could be associated with exacerbated glial activation [16]. Towards this end, we selected genes related to the GO terms “Extracellular matrix” (GO_0031012) and “Extracellular region” (GO_0005576) and analyzed their expression at 3 days post skull and nostril injury. Among all the regulated ECM-related transcripts (131), 69 of them were exclusively regulated after the skull injury (Figure 2D). These transcripts were overrepresented in GO terms related to the immune response, regulation of immune system process, and proteolysis (Figure 2E, Appendix A), processes implicated in the exacerbated glial reaction after wound closure. Moreover, some of these genes encoded for factors reported to regulate either glial reactivity (Ptpn6, Cst B, C1qa, C1qb, Mmp9, Fga) [62,63,64,65,66] or fibrosis [67,68,69,70,71] (Figure 2F). Because the two types of injuries show different kinetics in cellular response, some genes could still be differentially regulated at different time points after nostril injury. Therefore, we filtered out from the 1012 transcript set (Figure 2C) all transcripts regulated after nostril injury at any analyzed time point. We identified 812 transcripts regulated 3 days after skull injury but not at any time point after nostril injury (Figure 2G). Most of the GO terms significantly enriched in this gene set (2-fold enrichment and *p* < 0.01) were related to metabolism, immune, and innate immune response (Figure 2H, Appendix A).

In particular, we observed the upregulation of genes indicative of the activation of the Toll-like receptor, Tlr, (*mxc*, *mxe*, *irf7*, *irf2*) [72,73,74] and chemokine family 11 (*cxcl11.1*, *cxcl11.5*, *cxcl11.6like*, and *cxcl13*) [75] mediated innate immune response, at 3 days after skull injury (Figure 2I). Innate immunity orchestrates the initial events of wound healing after skin [76], heart [77], and CNS [9] injury, and its regulation determines the extent of tissue restoration [78]. Therefore, we set out to address whether the activation of either Tlr- or Cxcl11 family-mediated innate immunity leads to the induction of exacerbated glial reactivity in the zebrafish telencephalon. We first activated the Tlr-mediated innate immune response by injecting zymosan A microparticles [2] 3 days after nostril injury (Figure 2J) to mimic the temporal activation of this pathway observed after skull injury. Zymosan A was injected in the telencephalic ventricle and the glial reactivity was analyzed at 5 dpi, when the Olig2:GFP^+^ cell accumulation was already resolved after nostril and detected only after skull injury (Figure 2J,K,Q). Indeed, the vehicle (artificial cerebrospinal fluid, aCSF) treatment did not alter the reaction of Olig2:GFP^+^ cells and no accumulation was detected at 5 dpi (compare nostril 3 dpi Figure 1O with Figure 2Q for vehicle). In contrast, zymosan A treatment not only prolonged the accumulation of both 4C4^+^ and Olig2:GFP^+^ cells at the injury site (Figure 2K–M′,Q), but it also increased 7-fold the number of Olig2:GFP^+^ cells accumulating at the injury site 5 dpi compared with the vehicle treatment (Figure 2Q). Thus, zymosan A treatment turned the initial short-term glial activation into a prolonged and exacerbated accumulation of glial cells at the injury site after nostril injury. The toll-like receptor 2 (Tlr2) mediates the sterile inflammation induced by zymosan A in other systems [72,79], and Tlr2 was expressed in the intact as well as the injured zebrafish telencephalon (Appendix A). Therefore, we tested whether interfering with Tlr1/2 pathway activation using a Tlr1/2-specific competitive inhibitor (CU CPT22) would abolish the capacity of zymosan A to induce an exacerbated glial reaction after nostril injury (Appendix A). Indeed, interference with the activation of the Tlr1/2 pathways prevented the accumulation of Olig2:GFP^+^ cells at the injury site after zymosan A injection (Appendix A), suggesting that activation of Tlr1/2 is sufficient to induce a prolonged accumulation of Olig2:GFP^+^ cells and 4C4^+^ cells at the injury site.

Similar to Tlr2-induced innate immunity, we set out to test whether the Cxcl11 family has a role in prolonged glial activation, in line with the induction of these ligands exclusively after skull injury. As up-regulated Cxcl11-family ligands (Figure 2I) signal through the same chemokine receptor, Cxcr3 [80], we analyzed the ability of a specific Cxcr3 agonist (VUF 11222 [81]) to induce glial accumulation in the nostril injury paradigm (Figure 2N). Similar to the reactivity observed upon Tlr2 pathway activation (Figure 2K–M′,Q), treatment with the Cxcr3 agonist was sufficient to trigger exacerbated 4C4^+^ and Olig2:GFP^+^ cell accumulation at the injury site at 5 dpi (Figure 2O–Q).

Taken together, our data suggest that the activation of either Tlr2 or Cxcr3 is sufficient to induce an exacerbated glial reaction after nostril injury.

### 3.3. Tlr1/2 and Cxcr3 Pathways Cooperatively Control Reactive Gliosis after Injury in the Zebrafish Telencephalon

Because the activation of either Tlr2 or Cxcr3 signaling induced exacerbated glial reactivity in the nostril injury and the transcriptome analysis demonstrated the activation of both pathways exclusively after skull injury, we asked whether interference with these pathways would block the exacerbated gliosis after skull injury. We inhibited the activation of the two signaling pathways by using specific inhibitors: CU CPT22 for the Tlr1/2 [67] pathway and NBI-74330 for the Cxcr3 [68] pathway (Figure 3A). Strikingly, interference with the Tlr1/2 pathway did not change the accumulation of Olig2:GFP+ cells after skull injury (Figure 3A–C,F), despite a significant reduction in the area covered by 4C4+ immune cells (Appendix A). Likewise, the inhibition of the Cxcr3 pathway did not affect the accumulation of either 4C4+ or Olig2:GFP+ cells (Figure 3D,F and Appendix A). These data suggest that the two signaling pathways might be functionally redundant in controlling the accumulation of Olig2:GFP+ cells. To assess their redundancy, we simultaneously inhibited the Tlr1/2 and Cxcr3 pathways with respective inhibitors after skull injury (Figure 3A). Indeed, we observed a significant decrease in the number of Olig2:GFP+ cells accumulating at the injury site by 4 dpi in inhibitor-treated animals compared to vehicle (Figure 3E,F). Moreover, Sox10+ cells, representing oligodendrocyte lineage cells [69], showed a similar reduction, supporting the idea that the effect of inhibitors on the oligodendrocyte lineage was mainly in regulating their accumulation at the injury site, rather than affecting Olig2-driven expression of GFP (Figure 3B–E). In addition, the area covered by 4C4+ microglia/monocytes was also significantly reduced after double-inhibitor treatment (Appendix A). A reduction in the accumulation of Olig2:GFP+ cells at the injury site was also observed after Tlr1/2 inhibitor treatment (CU CPT22) combined with a different Cxcr3 inhibitor (AMG-48744) (Appendix A). Thus, the possibility of this phenotype being induced by the off-target effects of our pharmacological treatment is rather low. These effects of the inhibitor cocktail on alleviating Olig2:GFP+ glia and microglia/monocytes accumulation persisted also at later time points as no sign of Olig2:GFP+ cell accumulation was detectable following the double-inhibitor treatment 7 dpi in the skull injury paradigm (Figure 3G–J).

The reduction in the number of reactive glial cells accumulating at the injury site after double-inhibitor treatment suggests a role of these pathways in the initial induction of the glial cell reaction, their maintenance at the injury site, or both. To further disentangle the role of Tlr1/2 and Cxcr3 signaling in the maintenance of Olig2:GFP+ cells at the injury site, we pharmacologically blocked both pathways after the initial accumulation of Olig2:GFP+ cells at 4 dpi (Figure 3K). Once the glial cells had accumulated at the injury site (4 dpi), interference with the activation of both pathways failed to resolve the accumulation of Olig2:GFP+ cells 7 dpi (Figure 3L,M), in strong contrast to the improvement observed in the early inhibition protocol (Figure 3G,J). Taken together, these data support the role of Tlr1/2 and Cxcr3 signaling during the initial phase of glial accumulation.

Because immunohistochemical analysis showed a similar initial accumulation of glial cells at the injury site at 3 days following nostril and skull injury (Figure 1N,O), we asked whether interference with Tlr1/2 and Cxcr3 signaling could alter the accumulation of Olig2:GFP+ cells in the nostril injury paradigm. To address this question, we treated nostril-injured animals with Tlr1/2 and Cxcr3 inhibitors and assessed the accumulation of Olig2:GFP+ cells at the injury site (Appendix A). We observed a similar initial recruitment of Olig2:GFP+ at the injury site in untreated and vehicle-treated animals (Appendix A). Importantly, double-inhibitor treatment did not interfere with this initial accumulation of Olig2:GFP+ cells (Appendix A), in agreement with the absence of the transcriptional signature indicative of innate immunity activation after the nostril injury.

In conclusion, our data support the hypothesis that the restricted glial response correlating with complete tissue restoration and the long-lasting, reactive gliosis rely largely on different molecular mechanisms. The simultaneous activation of Tlr1/2 and Cxcr3 during the wound healing period is sufficient and necessary to induce a prolonged accumulation of both microglia/monocytes and Olig2:GFP+ cells at the injury site, leading to a long-lasting, exacerbated glial reaction.

### 3.4. Reduction in Glial Accumulation Correlates with Better Tissue Recovery

The reduction in the exacerbated accumulation of Olig2:GFP^+^ and microglia/monocytes after double-inhibitor treatment following skull injury prompted us to investigate the effect of prolonged injury-induced gliosis on brain regeneration by measuring the volume of the injured tissue (Figure 4A,B). We observed a significant reduction in the size of the injured tissue 7 dpi after double-inhibitor treatment compared with vehicle treatment (Figure 4B and Appendix A). This reduction in the injured volume was not observed in animals treated only with the Tlr1/2 pathway inhibitor (CU CPT22, Figure 4B) that maintains the Olig2:GFP^+^ cell accumulation but reduces microglial reactivity at 4 dpi (Appendix A). This finding supports the hypothesis that the decrease in the number of reactive Olig2:GFP^+^ cells at the injury site leads to improved tissue restoration.

We next tested whether the improved tissue recovery induced by the double-inhibitor treatment was accompanied by an addition of new, adult-generated HuC/D^+^ neurons to the injured brain parenchyma (restorative neurogenesis). As ependymoglial cells lining the ventricle surface increase their proliferation and generate new neurons in response to an injury [2,33,37], we used BrdU-based birth dating to determine whether the decreased glial reactivity after double-inhibitor treatment also correlated with improved restorative neurogenesis. To assess injury-induced neurogenesis, BrdU was added to the fish water during the first 3 days after injury to label all cells synthesizing DNA; that is, mostly dividing progenitors. The BrdU-incorporation phase was followed by a 4-day chase period, allowing progenitor differentiation, and correlating with the resolution of the glial accumulation upon inhibitor treatment (Figure 4C). We previously showed that the majority of newly generated neurons in the intact brain (BrdU^+^ and HuC/D^+^) reside in the ventricular zone (hemisphere periphery, Figure 4G) and display very low migratory potential [47]. Therefore, we analyzed the proportion of HuC/D^+^ and BrdU^+^ cells residing outside this neurogenic zone, as we observed the migration of new neurons towards this area only after injury [47]. Both control and inhibitor-treated animals generated similar total numbers of new neurons (HuC/D^+^ and BrdU^+^) after injury (Figure 4D–F). However, we observed a significantly increased proportion of new neurons located in the brain parenchyma after double-inhibitor treatment (HuC/D^+^ and BrdU^+^ located in the parenchyma in respect to all HuC/D^+^ and BrdU^+^ cells) (Figure 4G,H). As we did not observe any difference in the total number of generated neurons between control and double-inhibitor treated animals, our data exclude an effect of inhibitor treatment on injury-mediated stem cell activation, but rather support the interpretation that the resolution of a long-lasting, exacerbated glial reaction contributed to the better recruitment, survival, or integration of newly generated neurons into the injured brain parenchyma.

### 3.5. Microglia/Monocytes Depletion Does Not Alter the Innate Immunity-Regulated Accumulation of Olig2:GFP^+^ Cells at the Injury Site

The simultaneous inhibition of Tlr1/2 and Cxcr3 improved tissue regeneration. However, decreasing only 4C4^+^ cell reactivity with the Tlr1/2 inhibitor without changing Olig2:GFP^+^ cell accumulation showed no beneficial effect on infarct tissue volume (Figure 4B). These data suggest that microglia/monocytes might be unnecessary for the glial response regulated by the Tlr1/2 and Cxcr3 signaling pathways. To directly assess this hypothesis, we analyzed the accumulation of Olig2:GFP^+^ cells at the injury site in brains depleted of microglia/monocytes. A combination of Clodrosome and a Ccr2 inhibitor prior to skull injury depleted 95% of 4C4^+^ cells (microglia and infiltrating monocytes, Appendix A). The 4C4-free condition was then maintained by continuously blocking monocyte extravasation through Ccr2 inhibitor (Appendix A) during the restricted time window when the Tlr1/2 and Cxcr3 pathways induced the long lasting reaction of Olig2:GFP^+^ cells (Figure 2I and Figure 3A–F,K–M). Initial microglia/monocyte depletion did not alter Olig2:GFP^+^ cell accumulation at 4 days after skull injury compared with the control Encapsome treatment (Figure 4I,J,L and Appendix A). Importantly, the inhibition of Tlr1/2 and Cxcr3 successfully blocked the prolonged, exacerbated accumulation of Olig2:GFP^+^ cells in microglia/monocyte-depleted brains (Figure 4K,L) to the same extent these inhibitors prevent the accumulation of Olig2:GFP^+^ cells in brains populated with microglia/monocytes (compare double inhibitor in Figure 3F and Figure 4L. Consistent with this, our expression analysis of FACS-purified Olig2:DsRed^+^ cells (labeling the same oligodendroglia population as Olig2:GFP^+^) showed that they express Cxcr3 (Cxcr3.2 and Cxcr3.3) and Tlr2 (Tlr18) isoforms in both intact and injured brains (Appendix A). Moreover, the RNAscope analysis revealed the expression of genes involved in both innate immune pathways (*Cxcr3.2*, *Tlr8b*, *MYD88* and *Mxc*) in the Olig2:GFP^+^ population after skull injury (Appendix A). These data support the concept that the activation of microglia and/or invading monocytes is not necessary for Tlr1/2 and Cxcr3 injury-induced oligodendroglial reactivity and their initial accumulation at the injury site in zebrafish.

We next tested the effect of microglia/monocytes depletion on the restorative neurogenesis. To this end, we combined the depletion protocol with the BrdU-based neuronal birth dating used previously (Figure 4M). The initial depletion of injury-activated microglia/monocytes did not alter incorporation of new neurons (Figure 4N–O′) compared with untreated control animals (compare Veh in Figure 4H,R), supporting the hypothesis that the activated microglia/monocytes are not the only populations contributing to the adverse environment, restricting new neuron recruitment. Importantly, the inhibition of the Tlr1/2 and Cxcr3 pathways in microglia/monocyte-depleted brains still improved the addition of new neurons (Figure 4M–R), similar to the beneficial effects observed in animals with an intact immune system and further associating the beneficial effects of the double-inhibitor treatment with the resolution of prolonged Olig2:GFP^+^ cell accumulation.

Taken together, our results support the hypothesis that the Tlr1/2 and Cxcr3 pathways promote the accumulation of Olig2:GFP^+^ cells at the injury site and the injury-induced impairment of neuronal recruitment to the injury.

### 3.6. Olig2:dsRed^+^ Cells Activate Both Innate Immunity Pathways and Transcription Programs Involved in Cell Proliferation in Response to an Injury

In order to understand the regulatory mechanisms of accumulation of Olig2:GFP^+^ cells downstream of innate immunity pathways after skull injury, we analyzed the injury-induced transcriptomic changes in Olig2:dsRed^+^ cells (enriched for OPCs [82]) acutely isolated from the injured zebrafish telencephalon 3 days after either vehicle or inhibitor treatment. We observed 1649 significantly regulated transcripts in dsRed^+^ cells after injury in vehicle-treated brains compared with intact brains (Appendix A). Interestingly, a minority of transcripts were downregulated (114), suggesting that upon injury OPCs still maintain their oligodendrocyte lineage identity and gain additional features, leading to their reactivity. The distribution of upregulated genes in the biological pathways (Panther-based analysis) revealed the activation of FGF-, EGF-, PDGF-signaling pathways (Appendix A), which have previously been implicated in the proliferation of OPCs [83,84,85,86]. In line with those activated pathways, GO term analysis revealed an enrichment of the processes involved in reactive gliosis, such as cell migration and response to cytokines and chemokines (Appendix A). Surprisingly, most of the enriched GO terms were related to inflammation (63% of all enriched terms, Appendix A), including the activation of innate immunity. Importantly, the genes belonging to both cytokine and toll-like receptor signaling were upregulated in response to injury (Appendix A). Moreover, 45% of ECM- related genes specifically regulated at 3 days after skull injury in the entire telencephalon were also regulated in the Olig2:dsRed^+^ cell population (Appendix A). This unbiased transcriptome analysis further corroborated our hypothesis that cells of the oligodendrocyte lineage activate molecular pathways of the innate immune response, including Tlr2 and Cxcr3, which allows their microglia/monocyte-independent reaction and accumulation at the injury site.

The transcriptomic changes after skull injury, supporting the activation of innate immunity pathways directly in Olig2^+^ cells, prompted us to further analyze the effect of the inhibitor cocktail on gene expression in Olig2^+^ cells isolated from injured brains. Interestingly, the inhibitor treatment did not change the overall transcriptome of Olig2^+^ cells. Approximately 80% of regulated transcripts after inhibitor treatment were also regulated in vehicle-treated brains (Figure 5A and Appendix A). This suggests that the inhibitor cocktail treatment did not change the overall transcriptome of Olig2:dsRed^+^ cells, but rather restricted regulatory pathways involved in their long-term reactivity. Importantly, both cytokine receptor signaling and toll-like receptor signaling were no longer regulated in Olig2:dsRed^+^ cells after inhibitor treatment (Figure 5A,B; Appendix A). However, the regulation of number of biological processes linked to the immune response was still present (Appendix A). A comparison of injury-regulated genes in Olig2^+^ cells isolated from vehicle- and inhibitor cocktail-treated brains identified a set of 510 genes (597 transcripts) exclusively regulated after brain injury and vehicle treatment (Figure 5A) and, therefore, were likely involved in reactive gliosis downstream of the Tlr1/2 and Cxcr3 pathways. These genes were overrepresented in GO terms related to proliferation and cell migration (Figure 5B, Appendix A), both being biological processes at the core of the oligodendroglial reaction to injury and prolonged gliosis in mammals [87,88].

### 3.7. Regulation of Oligodendrocyte Progenitor Cell Proliferation by Tlr1/2 and Cxcr3 Signaling

Transcriptome data support the role of Tlr1/2 and Cxcr3 signaling pathways in the direct regulation of OPC proliferation leading to persistent accumulation at the injury site. Therefore, we first assessed the proliferation of Olig2:GFP+ cells after skull injury in the zebrafish telencephalon. Accordingly, we labeled all cells undergoing S-phase by BrdU within 5 days after the injury to find out if the proliferation contributes to the observed accumulation of oligodendroglia at the injury site (Figure 5C). As expected, we observed an accumulation of Olig2:GFP+ cells at the injury site, indicating that the BrdU treatment did not alter the behavior of Olig2:GFP+ cells. Importantly, we observed that 45% of all Olig2:GFP+ cells at the injury site were BrdU+ and hence went through at least one cell cycle during 5 days of labelling (Figure 5D–F), supporting the concept that the Olig2:GFP+ accumulation at the skull injury site was, at least in part, achieved by the increased proliferation of progenitor cells labelled by Olig2:GFP transgenic line (OPCs). We next analyzed if these accumulated OPCs further differentiated into mature oligodendrocytes. We made use of the transgenic line Tg(Mbp:nls-GFP) [77] and a BrdU-based birth dating protocol to identify the proportion of the injury-activated OPCs that matured into oligodendrocytes 7 days after the skull injury (Appendix A). We observed neither a significant increase in the total number of oligodendrocytes nor an increase in the proportion of newly matured, BrdU+ oligodendrocytes upon skull injury (Appendix A), supporting the concept that OPCs and not mature oligodendrocytes accumulate at the injury site [25].

Next, we analyzed whether the inhibitor cocktail treatment may alter the proliferation of Olig2:GFP+ cells as the cellular basis for reduction in their accumulation at the injury site. As Olig2:GFP+ cells display the first signs of exacerbated reactivity 3 days after skull injury, yet without the significant change in total number of Olig2:GFP+ cells, we analyzed the proliferation of Olig2:GFP+ cells 3 dpi after vehicle and inhibitor treatment (Figure 5G). This experiment revealed a significant reduction in the total number of BrdU+ Olig2:GFP+ cells after inhibitor cocktail treatment compared with the vehicle treatment (Figure 5H–L). To confirm the activation of Tlr1/2 and Cxcr3 pathways directly in OPCs, we made use of a murine OPC culture system. Moloney murine leukemia virus (MLV)-based clonal analysis was performed in pure primary OPC cultures isolated from P0 mouse cerebral cortex after vehicle or double-inhibitor treatment (Figure 5M). OPCs were permanently labeled with GFP expressing retrovirus and the size of clones produced by transduced progenitors within 5 days was measured (Figure 5N–P). Double-inhibitor treatment reduced the GFP+ clone size produced by OPCs, supporting a direct role of Tlr1/2 and Cxcr3 pathways in OPC proliferation (Figure 5P).

Taken together, our data indicate a direct role of Tlr1/2 and Cxcr3 pathways in regulating Olig2+ OPC proliferation to achieve long-lasting accumulation at the injury site in the zebrafish telencephalon.

### 3.8. Cerebrospinal Fluid Induces Exacerbated Glial Reactivity by Increasing OPC Proliferation

To identify the source and nature of the ligands activating the prolonged accumulation of OPCs after brain injury, we first examined the size of the skull versus nostril injury. As the volume of the skull injury was larger than the nostril injury (Figure 6A), we first set out to determine whether this was the cause of the reactive gliosis. We reduced the volume of the skull injury to one-third (small skull injury) using the same glass capillary as for the nostril injury (Figure 6B). The small skull injury still induced a strong reactivity of both 4C4+ and Olig2:GFP+ cells 7 days after the injury (Figure 6C,D). This reaction was comparable to the outcome of the initial skull injury, allowing us to exclude the size of the injury as a major determinant of differential glial reactivity.

We next hypothesized that an injury-induced ligand that activates the exacerbated reaction must be present only after skull injury. The telencephalic ventricle is located dorsally in the zebrafish brain [78,79] and, therefore, is exclusively damaged during the dorso-ventrally performed skull injury. Cerebrospinal fluid (CSF), which is confined to the ventricles, is rich in cytokines and growth factors that maintain normal homeostasis and nurture the brain; however, direct interaction with the brain parenchyma is restricted and regulated by the CSF–brain barrier [80]. Rupture of the ventricular barrier might allow an influx of CSF-derived molecules into the brain parenchyma, potentially explaining the activation of the Tlr1/2 and Cxcr3 pathways only after skull injury. To validate the potential of CSF to induced OPC reaction, we injected human CSF in the nostril injury site and analyzed glial reactivity (Figure 6E). Notably, we observed an 8-fold increase in the number of Olig2:GFP+ cells accumulating at the injury site (Figure 6F,M). As we inject the human CSF, the observed reaction could be a result of xenobiotic response. Therefore, we heat-inactivated the human CSF and probed its capacity to induce the reaction of OPCs in the nostril injury. Importantly, the dramatic CSF effect was not observed upon the administration heat-inactivated human CSF (Appendix A) Moreover, the administration of the human plasma containing many of the CSF components failed to induce the response (Appendix A), indicating that the prolonged OPC reactivity was not due to xenobiotic inflammation or misfolded proteins present in the CSF. The extraordinary potential of the CSF to induce exacerbated gliosis prompted us to investigate the cellular basis for the Olig2:GFP+ cell accumulation in response to the CSF. As the accumulation of OPCs after a skull injury was achieved, at least in part, by an increased proliferation of OPCs (Figure 6C–F), we assessed whether the proliferation of Sox10+ cells was also induced by the CSF injection into the nostril injury site (Figure 6G). Indeed, we observed that the majority of Sox10+ cells accumulating around the injury site incorporated BrdU during the initial 3 days after the injury and CSF administration (Sox10+ and BrdU+ cells in respect to all Sox10 + cells) (Figure 6H,H′,J). The induced proliferation was not, however, observed after the injection of heat-inactivated CSF (Figure 6I,J), in line with the significantly smaller accumulation of Olig2:GFP+ cells at the injury site observed after heat-inactivated CSF treatment (Appendix A). The similarity in OPC reaction induced by CSF injection into the nostril injury and the skull injury motivated us to assess whether CSF-induced accumulation of OPCs involved the activation of the Tlr1/2 and Cxcr3 pathways. Therefore, we inhibited the Tlr1/2 and Cxcr3 pathways together with the administration of human CSF after nostril injury (Figure 6K). Importantly, the accumulation of Olig2:GFP+ cells was prevented upon Cxcr3 and Tlr1/2 inhibition, despite the accessibility of the CSF at the injury site (Figure 6L,M). Taken together, these data suggest that the OPC accumulation observed upon skull injury is likely triggered by leakage of CSF into the brain parenchyma and the subsequent activation of the Tlr1/2 and Cxcr3 pathways.

To identify potential ligands activating innate immunity pathways in the CSF, we set up an in vitro system that relays on the proliferation of a murine OPC cell line (OliNeu). Importantly, the addition of CSF to the OliNeu culture medium induced a dose-dependent increase in the proportion of proliferating, phospho-histone H3 (pH3) positive cells (Figure 7A–D), in line with our data that human CSF can directly regulate OPC proliferation in vivo (Figure 6J). Moreover, this dose-dependent response was completely abolished in the double Tlr2 and Cxcr3 knockout clones generated using CRISPR-Cas9 technology (Figure 7D,Q and Appendix A). These results not only confirmed the pivotal role of Tlr2 and Cxcr3 signaling in the CSF induced proliferation of OPCs, but also additionally validated the specificity of our pharmacological inhibitor treatment in vivo.

As cytokines have been reported to activate both Cxcr3 and Tlr2 signaling [61,65,81], we first studied the composition of four healthy donor-derived CSFs using a cytokine antibody array (Figure 7E,F). It is important to mention that all four samples increased proliferation in vivo. Strikingly, 90% of the analyzed cytokines were present in at least one of the samples and 57% in all four samples (Figure 7E,F).

Out of these cytokines, we pre-selected 30 potential candidates (Figure 7E; Appendix A) that were present in at least one CSF sample and available as recombinant protein for further functional screening using OliNeu proliferation as a read-out (Figure 7G). We used three different concentrations of the selected candidates and six of them (Ccl5, EGF, Ccl7, IL-10, Cxcl9 and IL-3) significantly increased the proportion of mitotic pH3+ cells (Figure 7G), including a known Cxcr3 ligand (Cxcl9). Interestingly, some candidates such as Cxcl9 were not detected in all CSF samples despite the ability of all four CSF samples to induce the accumulation of Olig2:GFP+ cells, suggesting redundancy of ligands in their capacity to activate OPC proliferation. Taken together, our data suggest that the CSF cytokines activate the innate immunity to regulate OPC proliferation in a redundant manner and therefore regulate the reactive gliosis.

## 4. Discussion

Despite the general agreement that mammals exhibit a limited regenerative capacity after CNS trauma, it remains controversial which specific cellular and molecular mechanisms trigger the long-lasting glial reaction that in turn, negatively impact the endogenous regeneration. Comparative studies analyzing the regeneration of competent and incompetent species have failed to identify the specific mechanisms involved in the pathogenesis of traumatic injuries [89,90,91,92,93,94], even after comparing evolutionarily close species [95,96,97]. This is due to the complexity of the wound healing response that involves a number of molecular pathways and different cell types [9,34]. There has been a long-held belief that microglia and astrocytes are at the core of the poor regenerative outcome [98,99]. However, recent studies have challenged this concept [5,14]. Ablation of microglia upon CNS injury failed to improve neuronal survival and functional recovery and in some cases, even worsened the regenerative outcome [100,101]. Astrocytic activation after trauma appeared to be more complex than originally expected. Microglia-induced inflammation regulates the activation of different types of reactive astrocytes named “A1” and “A2” [18]. While A1 displayed a neurotoxic phenotype, A2 astrocytes appeared to exert neuroprotective functions [64]. The differential activation of A1 and/or A2 astrocytes might explain the controversy about the functional consequences of astrogliosis and whether reactive astrocytes promote endogenous regeneration or contribute to the detrimental reactive gliosis. Surprisingly, and despite their rapid and robust reaction to injury, it is largely unknown how the oligodendrocyte progenitors (OPCs) fit in this inflammatory cascade. To study the role of OPCs during the wound healing process, we performed a comparative analysis of two injury paradigms displaying differential OPC reactivity in the same organ and model organism. In contrast to the previously described nostril injury of brain parenchyma [33,37,38], skull injury showed more similarities to the glial response reported in mammals, such as prolonged OPC accumulation, lack of tissue restoration, extracellular matrix modifications, and exacerbated inflammatory response [34,102].

Comparative analysis (nostril vs. skull) revealed a shared cellular reaction and large overlap in gene regulation shortly after injury, highlighting the common features of the initial wound healing processes regardless if it is associated with prolonged, exacerbated, or restricted glial reactivity [9]. Importantly, a unique molecular signature, including specific innate immunity pathways, was expressed 3 days after the skull injury, correlating with the first signs of the exacerbated glial reaction. These pathways were never regulated during the nostril wound healing, validating the theory that the prolonged, reactive gliosis is induced by a specific molecular program independent of the initial wound healing response [19]. Our study does not support the concept that the regenerative capacity of the CNS is an evolutionarily fixed feature of a given species [103,104]; rather, it is a highly regulated, adaptive response to a specific type of injury. This concept is shared with skin regeneration, in which the depth of the injury determines the scar response [105].

We identified two receptors, Tlr1/2 and Cxcr3, as main regulators of the exacerbated glial reactivity. Interestingly, the inhibition of either of the two pathways separately showed no beneficial effect, whereas the activation of either Tlr2 or Cxcr3 in the nostril paradigm was sufficient to induce gliosis. Hence, both signaling pathways control reactive gliosis in a redundant and synergistic manner. As previously discussed, the complex cross-regulation of immune cells (monocytes and microglia) and astrocytes after brain injury is crucial for the regenerative outcome [19,106,107,108]. The classical inflammatory cascade is initiated by activation and polarization of microglia and invading monocytes. Consequently, these cells regulate the reactivity of astrocytes that, in turn, limits the inflammatory response [17,19,109,110]. However, our data demonstrate, for the first time, that microglia/monocytes are not essential for the initial activation of OPCs and that the Tlr1/2 and Cxcr3 pathways can be directly regulated in this population. Our study, therefore, brings forth a new concept that OPCs can sense and react to injury-induced signals independent of microglia and invading monocytes. However, we cannot exclude any involvement of microglia/monocytes in other aspects of the wound healing. As we did not perform cell-specific interference and because both receptors are expressed in several cell types (microglia/monocytes [111,112]; astrocytes [113,114]; oligodendroglia [114]; neurons [113,115]), we cannot exclude the possibility that other cell types contribute to the induction of the wound closure via the Tlr1/2 and Cxcr3 signaling pathways. However, our knockout in vitro model validated the role of the Tlr2 and Cxcr3 pathways in directly activating OPC proliferation and hence, demonstrated that they are clearly involved in a crucial manner in the reactive gliosis. Moreover, the improvement in tissue recovery (reduced injured volume and enhanced restorative neurogenesis) observed after double-inhibitor treatment correlates nicely with a reduction in the number of accumulating Olig2^+^ cells. Our data support the hypothesis that the detrimental environment classically associated with the reactive gliosis might be driven specifically by reactive OPCs.

The central role of the Tlr1/2 and Cxcr3 pathways in regulating gliosis and tissue restoration motivated us to investigate injury-induced mechanisms. Our study suggests that the ligand(s) activating the Tlr1/2 and Cxcr3 pathways are part of the CSF that leaks into the CNS parenchyma upon traumatic injury [116]. The capability of the CSF to directly induce OPC proliferation in a Tlr2 and Cxcr3-dependent manner, further corroborates the pivotal role of OPCs in initiating the long-lasting glial response and generating a harmful environment. Although the specific CSF-derived molecule/s driving the reactive gliosis in vivo remain unidentified, our in vitro screening suggests that some of the cytokines present in the CSF could be responsible for the OPC activation induced by the traumatic brain injury. These data support a central regulatory role of CSF in controlling not only the activation of neural stem cells in the intact brain, but also the activation state of CNS glia after injury [117,118]. Overall, our work highlights novel pathways in exacerbated OPC activation as potential targets for developing efficient therapies improving regeneration in the mammalian brain.

## Figures and Tables

**Figure 1 cells-11-00520-f001:**
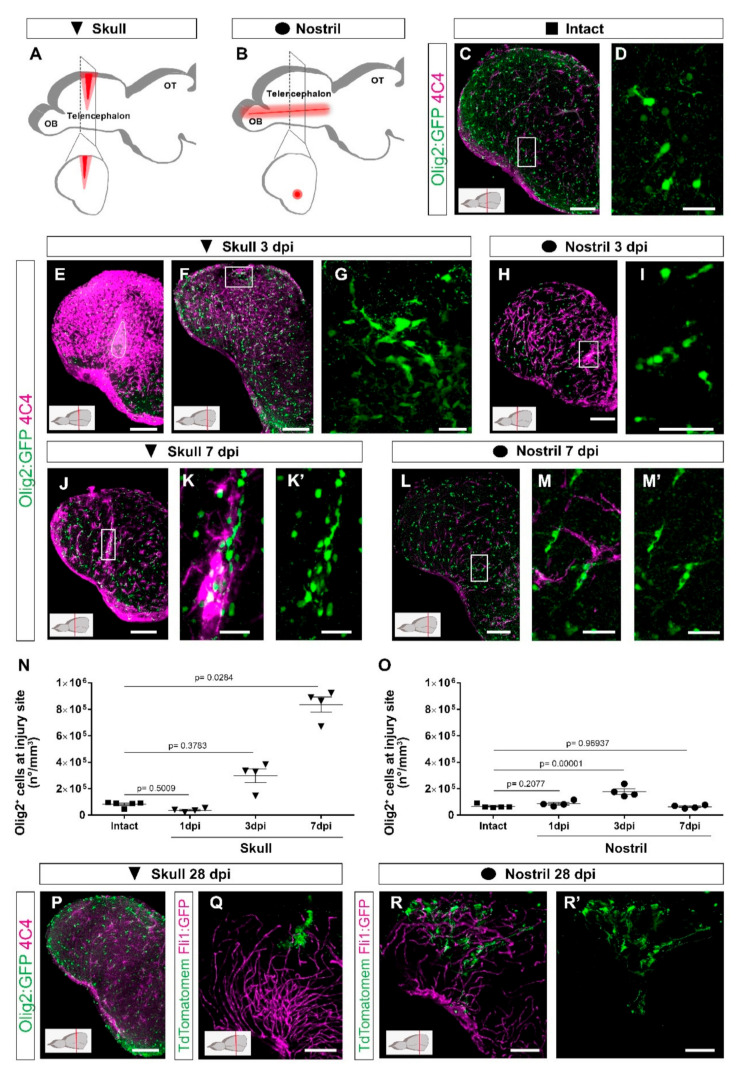
Distinct injury paradigms in the zebrafish telencephalon led to either scarless regeneration or prolonged glial reactivity. (**A**,**B**) Schemes depicting skull (**A**) and nostril (**B**) injury paradigms. Red triangle (**A**) and red line (**B**) illustrate the injury track. (**C**,**D**) Micrographs of a telencephalic section showing the distribution of Olig2:GFP^+^ oligodendroglia and 4C4^+^ microglia/monocytes in the intact brain. (**E**,**F**) Images of 3 dpi skull-injured sections (4C4^+^ and Olig2:GFP^+^ cells) at the level of the injury core delineated by a white line (**E**) and lateral to the injury core depicting the first signs of Olig2:GFP^+^ cells to accumulation indicated by the boxed area (**F**). (**H**) Image showing the distribution of Olig2:GFP^+^ and 4C4^+^ cells at 3 dpi after a nostril injury. (**G**,**I**) are magnifications of the boxed areas in (**F**) and (**H**), depicting Olig2:GFP^+^ cell distribution. (**J**–**M′**) Images showing the reactivity of 4C4^+^ and Olig2:GFP^+^ cells at 7 days after skull (**J**) and nostril (**L**) injury. (**K**,**K′**,**M**,**M′**) are magnifications of the boxed area in the respective images. (**N**,**O**) Graphs depicting the density of Olig2:GFP^+^ cells at the injury site after skull (**N**) and nostril (**O**) injury. Data are shown as mean ± SEM; each data point represents one animal. Statistical analysis is based on a non-parametric Kruskal–Wallis Test (*p*-value = 0.0021) with a post-hoc Dunn test (Many-to-One) in **N** and a one-way ANOVA (*p*-value = 2.483 × 10^−5^) with a post-hoc Dunnett test (many-to-one) in (**O**). (**P**) The accumulation of 4C4^+^ and Olig2:GFP^+^ cells resolved at 28 days after skull injury. (**Q**–**R′**) Images showing the morphology of ependymoglial cells (labelled by electroporation of TdTomatomem) 28 days after skull (**Q**) and nostril injury (**R**,**R′**). While we observed the restoration of the radial morphology of the labelled ependymoglia that contacts the basement membrane after nostril injury (similar to the intact brain), the ependymoglia after skull injury failed to restore radial morphology and built extensive contacts with Fli1-positive blood vessels. All images are full z-projections of a confocal stack; insets indicate the rostro-caudal levels of the sections. Scale bars in (**C**,**E**,**F**,**H**,**J**,**L**,**P**,**Q**,**R**,**R′**) = 100 µm; Scale bars in (**D**,**G**,**I**,**K**,**K′**,**M**,**M′**) = 20 µm. Abbreviations: OB: olfactory bulb, OT: optic tectum, dpi: days post-injury; AFOG: acid fuchsin orange G. Symbol description: black triangle: skull injury; black circle: nostril injury.

**Figure 2 cells-11-00520-f002:**
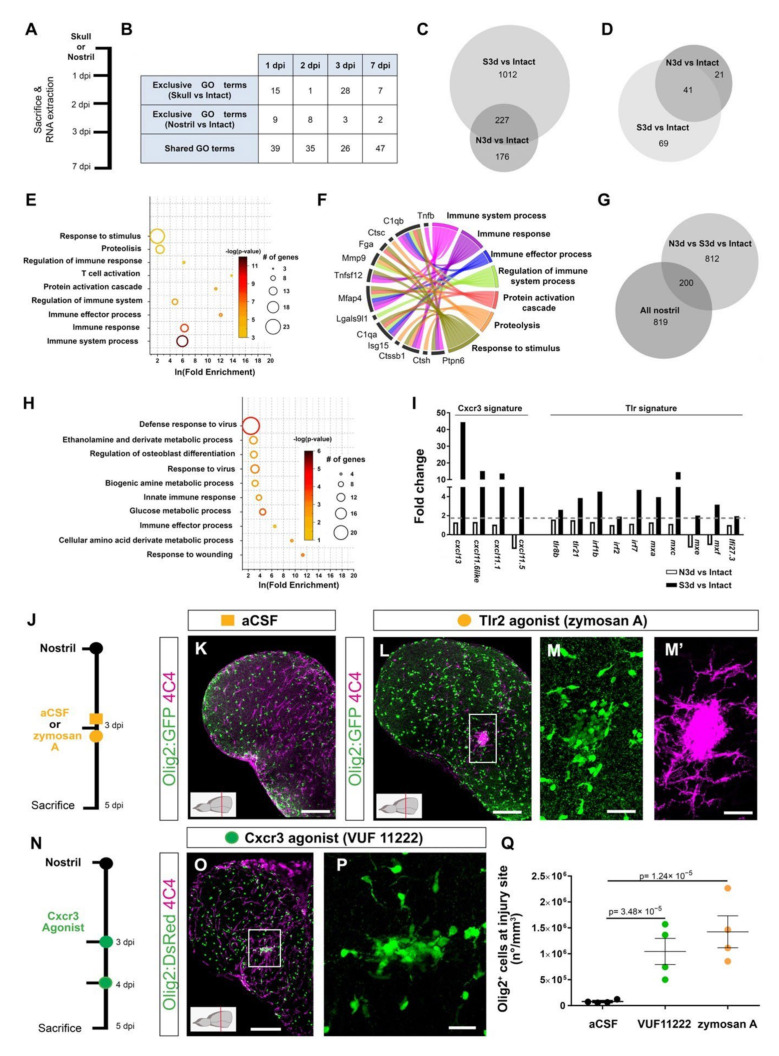
Activation of innate immunity pathways after injury induced a prolonged glial reaction in the zebrafish telencephalon. (**A**) Experimental design to analyze transcriptome changes occurring upon nostril and skull injury. (**B**) Table depicting the number of significantly regulated Gene Ontology terms (Injury vs. Intact) at different time points after nostril and skull injury. (**C**) Comparative analysis using a Venn diagram illustrating the number of genes exclusively regulated 3 days after skull injury (Skull vs. Intact) and not after nostril 3 dpi (Nostril vs. Intact). (**D**) Venn diagram depicting the overlap between ECM-related genes regulated at 3 days after skull and nostril injury. Regulated genes were defined by a *p*-value < 0.05, fold-change > 1.6, and a linear average expression > 20. (**E**) Significantly enriched Gene Ontology (GO) terms of biological processes (color indicates *p*-values and symbol size number of identified genes within the term) in an ECM-related gene set regulated exclusively 3 days after skull injury (69 genes in panel (**D**)). (**F**) Chord diagram depicting selection of regulated ECM-related genes and associated GO terms biological processes. (**G**) Venn diagram depicting the overlap between genes exclusively regulated at 3 days after skull injury and genes regulated after the nostril injury at any time point. Note that 80% of the genes were exclusively regulated after skull injury at 3 dpi but were never regulated after nostril injury. (**H**) Plot showing significantly enriched (*p*-values indicated on bars) GO terms related to biological processes in a gene set regulated exclusively 3 days after skull injury (Skull 3 dpi vs. Nostril at any time point), correlating with glial accumulation. (**I**) Histogram depicting the regulation of genes related to Cxcr3 and Tlr signatures after nostril and skull injury. The dotted, gray line shows the 1.6-fold change cut off. (**J**) Scheme of the experimental design analyzing the ability of the Tlr2 agonist to induce glia accumulation after nostril injury. (**K**,**L**) Images of 5-day-injured telencephalic sections in the *Tg(Olig2:GFP)* line after nostril injury and aCSF (**K**) or zymosan A injections (**L**). (**M**,**M′**) Magnifications of the boxed area in L depict the exacerbated accumulation of Olig2:GFP^+^ (**M**) and 4C4^+^ (**M′**) cells at the injury site. (**N**) Scheme representing the experimental design to analyze the capacity of the Cxcr3 agonist (VUF 11222) to induce a reactive gliosis. (**O**) Micrograph illustrating the reactivity of Olig2:DsRed^+^ and 4C4^+^ cells after Cxcr3 activation. (**P**) Magnification of the injured area in (**O**). (**Q**) Graph showing the density of Olig2:GFP^+^ cells in the injured area 5 days after nostril injury with aCSF, Cxcr3 or Tlr2 agonist treatments. Data are shown as mean ± SEM; each data point represents one animal. *p*-values are based on a one-way ANOVA (*p*-value = 1.183 × 10^−5^) with a post-hoc Dunnett test (Many-to-One). All images are full z-projections of a confocal stack. Insets indicate the rostro-caudal levels of the sections. Scale bars in (**K**,**L**,**O**) = 100 µm; scale bars in (**M**,**M′**,**P**) = 20 µm; Abbreviations: dpi: days post-injury, N3d: nostril 3 dpi, S3d: skull 3 dpi; Ctrl: control; aCSF: artificial cerebrospinal fluid. Symbol description: orange square: ventricular injection of aCSF; orange circle: ventricular injection of zymosan A, Tlr2 agonist; green circle: VUF 11222, Cxcr3 agonist; black circle: nostril injury.

**Figure 3 cells-11-00520-f003:**
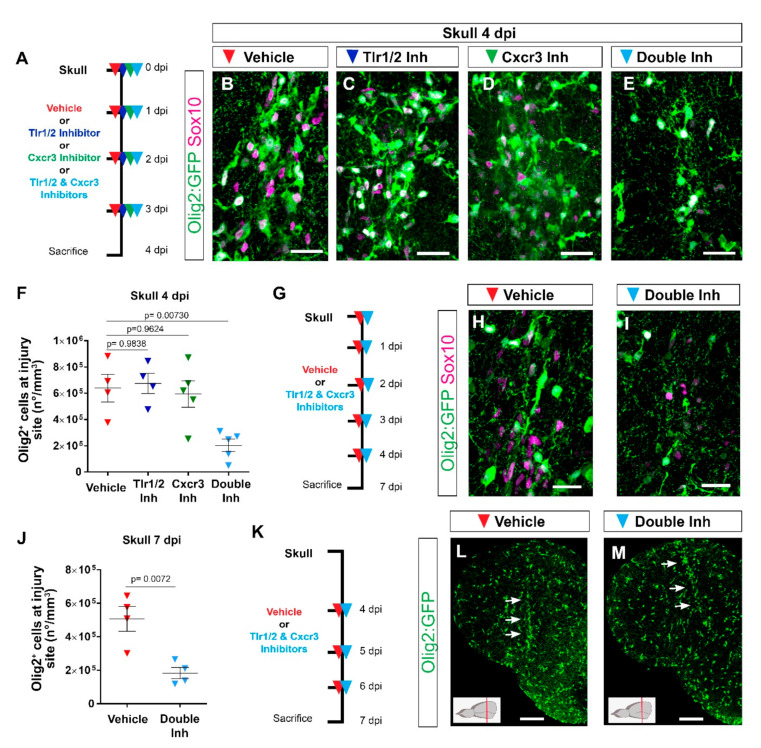
Tlr1/2 and Cxcr3 pathways redundantly control the accumulation of Olig2:GFP^+^ cells but not their maintenance at the injury site in the zebrafish telencephalon. (**A**) Scheme of the experimental setup to address the role of Cxcr3 and Tlr1/2 in the reactive gliosis 4 days after injury. (**B**–**E**) Micrographs of telencephalic sections obtained after 4 dpi depicting Olig2:GFP^+^ and Sox10^+^ oligodendroglia reactivity with vehicle (**B**), Tlr1/2 inhibitor (**C**), Cxcr3 inhibitor (**D**), and double-inhibitor (**E**) treatments. (**F**) Graph showing the density of Olig2:GFP^+^ cells located at the injury site after vehicle, Tlr1/2 inhibitor (CU CPT22), Cxcr3 inhibitor (NBI 74330) and double-inhibitor combination (NBI 74330 + CU CPT22) treatment. Note that only the double-inhibitor cocktail reduces the number of Olig2:GFP^+^ cells accumulating at the injury site. Data shown as mean ± SEM; each data point represents one animal. *p*-values are based on a one-way ANOVA (*p*-value = 4.074 × 10^−3^) with a post-hoc Dunnett test (Many-to-One). (**G**) Experimental outline to assess the effect of vehicle and double-inhibitor treatment 7 dpi. (**H**,**I**) Micrographs of telencephalic sections 7 days after skull injury depicting Olig2:GFP^+^ and Sox10^+^ oligodendroglia after vehicle (**H**) and inhibitor cocktail (NBI 74330 and CU CPT22) (**I**) treatment. (**J**) Graph illustrating the density of Olig2:GFP^+^ cells located within the injured volume after vehicle and double-inhibitor treatment. An equal volume was quantified in both conditions (*p*-values is based on Student’s *t*-test with equal variances). (**K**) Scheme depicting the experimental design to assess the capacity of the vehicle and double inhibitors treatment to resolve glial accumulation. (**L**,**M**) Micrographs showing telencephalic sections 7 days after skull injury and vehicle (**L**) or double-inhibitor (**M**) treatment. White arrows indicate the injury site. Note that both vehicle and inhibitor treatments failed to resolve Olig2:GFP^+^ accumulation. All images are full z-projections of confocal stack. The level of the cross-section is indicated in the inset. Scale bars in (**L**,**M**) = 100 µm; scale bars in (**B**–**E**,**H**,**I**) = 20 µm. Abbreviations: dpi: days post-injury, Inh: inhibitor. Symbol description: red triangle: vehicle; dark blue triangle: Tlr1/2 inhibitor, CU CPT22; green triangle: Cxcr3 inhibitor, NBI 74330; light blue triangle: double inhibitors, NBI 74330 and CU CPT22.

**Figure 4 cells-11-00520-f004:**
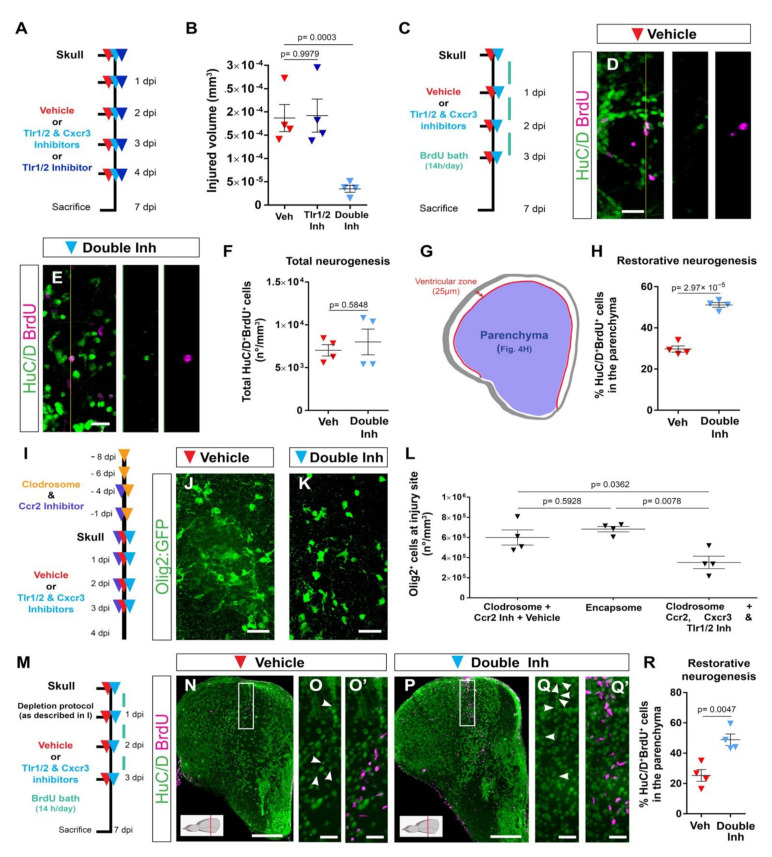
Activation of Tlr1/2 and Cxcr3 creates a detrimental environment by inducing oligodendroglia accumulation in a microglia/monocyte-independent manner. (**A**) Scheme of the experimental design to analyze the consequences of double-inhibitor treatment (NBI 74330 and CU CPT22). (**B**) Graph illustrating the size of the injured volume 7 days after skull injury and vehicle, Tlr1/2 inhibitor (CU CPT22), or double Tlr1/2 and Cxcr3 inhibitor (NBI 74330 and CU CPT22) treatment. *p*-values are based on a one-way ANOVA (*p*-value = 1.971 × 10^−4^) with a post-hoc Dunnett test (Many-to-One). (**C**) Experimental scheme designed to study restorative neurogenesis upon different treatments. (**D**,**E**) Images depicting HuC/D^+^ and BrdU^+^ cells located in the parenchyma following vehicle (**D**) and double-inhibitor (**E**) treatment. (**F**) Dot-plot showing the total density (whole telencephalon) of HuC/D^+^ and BrdU^+^ after vehicle and Tlr1/2 and Cxcr3 inhibitor treatment. *p*-value is based on WelcH′s *t*-test with unequal variances. (**G**) Diagram illustrating the ventricular zone (25 µm from the ventricle surface) and the parenchyma (blue area) in the telencephalic region. Restorative neurogenesis was measured by the proportion of newly generated neurons (HuC/D^+^ and BrdU^+^) that migrated towards the parenchyma with respect to the total number (ventricular zone and parenchyma) of new neurons. (**H**) Graph depicting the proportion of HuC/D^+^ and BrdU^+^ cells located in the telencephalic parenchyma after vehicle and Tlr1/2 and Cxcr3 inhibitor treatment. *p*-value is based on Student’s *t*-test with equal variances. (**I**) Design of the experimental workflow to analyze the effect of Tlr1/2 and Cxcr3 inhibitors on accumulation of Olig2:GFP^+^ cells after microglia/monocytes depletion. (**J**,**K**) Micrographs depicting the reactivity of Olig2:GFP^+^ cells after skull injury at 4 dpi with microglia/monocyte depletion and vehicle (**J**) or Tlr1/2 and Cxcr3 inhibitor treatments (**K**). (**L**) Graph illustrating the density of Olig2:GFP^+^ cells at the injury site at 4 dpi following Clodrosome + Ccr2 (MK-0812) inhibitor treatment (microglia/monocyte depletion protocol), Encapsome (empty liposomes, control for Clodrosome; ventricular injection) and Clodrosome + Ccr2 + Tlr1/2 (CU CPT22) + Cxcr3 (NBI 74330) inhibitor treatments. The decrease in Olig2:GFP^+^ cell accumulation after Tlr1/2 and Cxcr3 inhibitor treatment was maintained in microglia/monocyte-depleted brain. *p*-values are based on a one-way ANOVA (*p*-value = 7.957 × 10^−3^) with a post-hoc Tukey Test (All Pairs). (**M**) Design of the experimental protocol used to analyze injury-induced neurogenesis (BrdU-based birth dating) in microglia/monocyte-depleted brains treated with vehicle or Tlr1/2 and Cxcr3 inhibitor cocktail. (**N**,**P**) Micrographs of injured telencephala at 7 dpi showing the generation of new neurons (HuC/D^+^/BrdU^+^) after vehicle (**N**) and Tlr1/2 and Cxcr3 inhibitor (**P**) treatment in microglia/monocyte-depleted brains. (**O**,**O′**,**Q**,**Q′**) are magnifications of the areas boxed in (**N**,**P**), respectively. White arrowheads depict double HuC/D^+^ and BrdU^+^ cells. The level of the cross-section is indicated in the inset. (**R**) Graph depicting the proportion of HuC/D^+^ and BrdU^+^ cells located in the telencephalic parenchyma after vehicle and Tlr1/2 and Cxcr3 inhibitor treatment. *p*-value is based on Student’s *t*-test with equal variances. All images are full z-projections of a confocal stack. Data are shown as mean ± SEM; each data point represents one animal. Scale bars in (**N**,**P**) = 100 μm; scale bars in (**D**,**E**,**J**,**K**,**O**,**O′**,**Q**,**Q′**) = 20 μm. Abbreviations: dpi: days post-injury; Veh: vehicle; Inh: inhibitors. Symbol description: red triangle: vehicle; dark blue triangle: Tlr1/2 inhibitor, CU CPT22; light blue triangle: double inhibitors, NBI 74330 and CU CPT22; orange triangle: ventricular Clodrosome injection; purple triangle: intraperitoneal Ccr2 inhibitor injection, MK-0812.

**Figure 5 cells-11-00520-f005:**
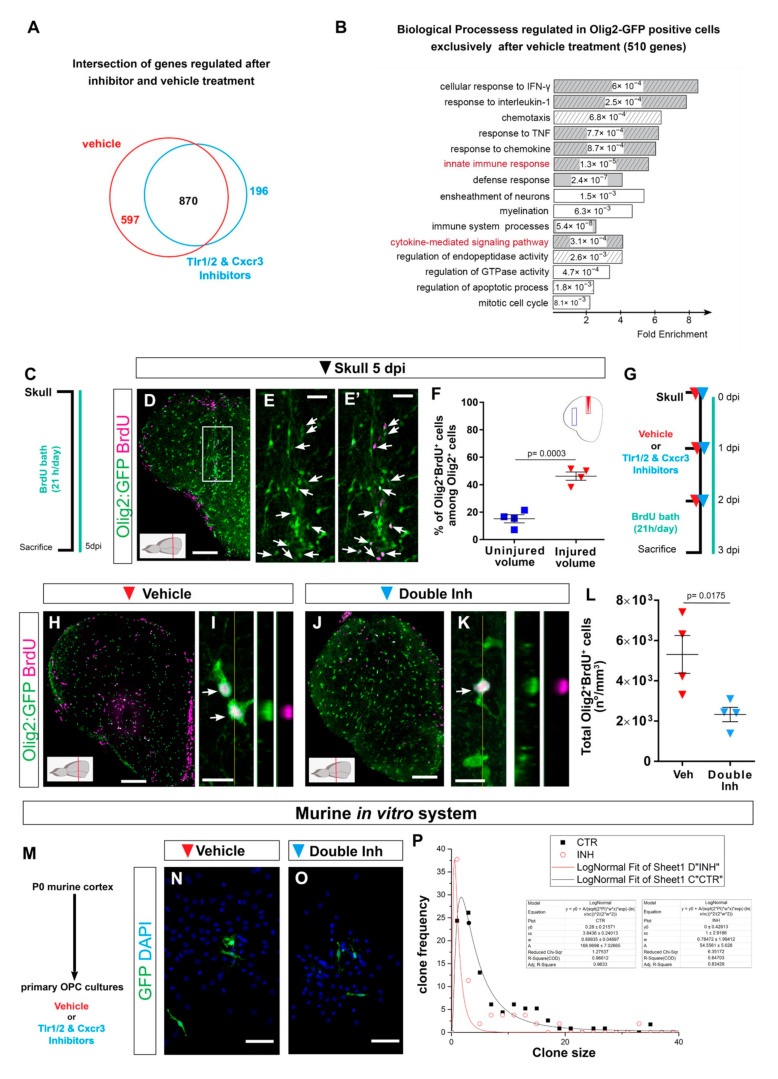
Transcriptome analysis of zebrafish oligodendrocyte lineage reveals the activation of innate immunity and cell cycle pathways after skull injury. (**A**) Venn diagram of genes regulated at 3 dpi in Olig2-GFP^+^ cells after vehicle (red) and Tlr1/2 and Cxcr3 inhibitor (cyan) treatment. (**B**) Histogram depicting GO biological process terms significantly enriched (*p*-values indicated on bars) in a gene set (597 genes in (**A**)) normalized after inhibitor treatment and therefore regulated exclusively after vehicle treatment. GO terms related to inflammatory response are shown by gray bars; patterned bars indicate processes previously reported to be activated in response to injury. Note that both innate immunity and cytokine-mediated signaling pathways are normalized upon inhibitor treatment. (**C**) Scheme depicting the experimental design to analyze the proliferative capacity of Olig2:GFP^+^ cells during the first 5 days after skull injury. (**D**) Micrograph of injured section 5 days after skull injury stained for GFP and BrdU. (**E**,**E′**) Magnification of the oligodendroglial accumulation boxed in (**D**). Double Olig2:GFP^+^ and BrdU^+^ cells are marked with white arrows. (**F**) Graph illustrating the proportion of Olig2:GFP^+^ and BrdU^+^ cells located at the injury site and in an equivalent uninjured volume in the same section. Note that 45% of the Olig2:GFP^+^ cells at the injury site proliferated after skull injury. (**G**) Scheme of the experimental design to assess the proliferation of Olig2-GFP^+^ cells after vehicle and inhibitors treatment. (**H**,**J**) Images of telencephalic sections 3 days after skull injury and BrdU bath with vehicle (**H**) and double inhibitors (**J**) treatments. (**I**,**K**) Micrographs with orthogonal projections of proliferating (BrdU^+^) Olig2:GFP^+^ cells after vehicle (**I**) and Tlr1/2 and Cxcr3 inhibitor (**K**) treatment. (**L**) Graph depicting the density of Olig2:GFP^+^ and BrdU^+^ cells 3 dpi in vehicle and Tlr1/2 and Cxcr3 inhibitor treated animals. (**M**) Experimental design to measure the clonal growth of murine OPCs primary cultures after vehicle and Tlr1/2 and Cxcr3 inhibitor cocktail treatment. OPCs were permanently labeled with GFP expressing retrovirus. (**N**,**O**) Micrographs depicting OPC derived clones 5 days after retroviral infection in vehicle (**N**) and Tlr1/2 and Cxcr3 inhibitor (NBI 74330 and CU CPT22) cocktail (**O**) treated primary OPCs culture. (**P**) Graph depicting the frequency of different clone sizes in the vehicle (CTR) and Tlr1/2 and Cxcr3 inhibitor cocktail (INH) treated primary OPCs culture. Data are shown as mean ± SEM; each data point represents one animal. *p*-values are based on Student’s *t*-test with equal variances. All images are full z-projections of a confocal stack. The level of the cross-section is indicated in the inset. Scale bars in (**D**,**H**,**J**) = 100 μm; scale bars in (**N**,**O**) = 50μm, scale bars in (**E**,**E′**) = 20 μm; scale bars in (**I**,**K**) = 10 μm. Abbreviations: dpi: days post-injury; Veh: vehicle; Inh: inhibitors; OPC: oligodendrocyte progenitor cell. Symbol description: Triangle: skull injury; blue square: uninjured volume; red triangle: vehicle; light blue triangle: double inhibitors, NBI 74330 and CU CPT22; black square: control primary OPCs; red circle: double inhibitor (NBI 74330 and CU CPT22) treated primary OPCs.

**Figure 6 cells-11-00520-f006:**
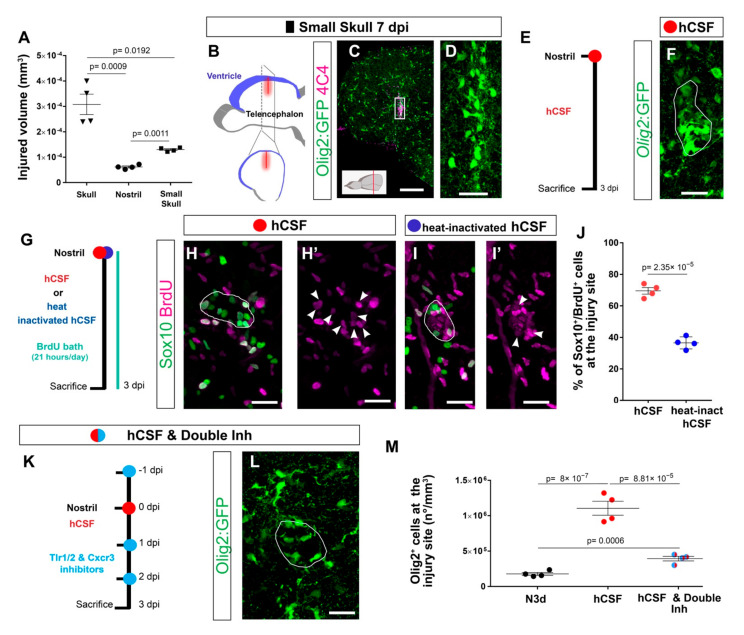
Cerebrospinal fluid-derived molecules induce the proliferation of OPCs and a reactive gliosis-like phenotype. (**A**) Graph depicting the size of the injured volume after skull, nostril, and small skull injury at 1 dpi. *p*-values are based on a Welch one-way ANOVA (unequal variances; *p*-value = 3.034 × 10^−4^) with a post-hoc Dunnett T3 test with unequal variances (all pairs). (**B**) Scheme depicting the small skull injury model. Nostril and small skull injuries were performed with a glass capillary. The red line indicates the dorso-ventral injury through the skull and blue indicates the location of the telencephalic ventricle. (**C**) Image illustrating the reactivity of Olig2:GFP^+^ and 4C4^+^ cells 7 days after small skull injury. (**D**) Magnification of the oligodendroglial accumulation boxed in (**C**). (**E**) Design of the experimental workflow to analyze the effect of human CSF administration. (**F**) Image illustrating the reactivity of Olig2:GFP^+^ cells 3 days after nostril injury and hCSF treatment. White line depicts the injury site. (**G**) Experimental design to analyze the proliferative capacity (BrdU incorporation) of Sox10^+^ cells after nostril injury at 3 dpi and hCSF or heat-inactivated hCSF administration. (**H**–**I′**) Images showing the accumulation of Sox10^+^ and BrdU^+^ cells at the nostril injury site after hCSF (**H**,**H′**) or heat-inactivated human CSF (**I**,**I′**) administration. White lines depict the injury site and white arrowheads the colocalization of BrdU and Sox10. (**J**) Dot-plot depicting the proportion of Sox10^+^ and BrdU^+^ cells accumulating at the nostril injury site after hCSF or heat-inactivated hCSF administration. *p*-value is based on Student’s *t*-test with equal variances. (**K**) Workflow to study the effect of the Tlr1/2 and Cxcr3 inhibitor treatment after human CSF injection. (**L**) Micrograph of a nostril-injured telencephalon at 3 dpi depicting Olig2:GFP^+^ cell reactivity following human CSF and inhibitor treatment. The white line depicts the injury site. (**M**) Graph showing the density of Olig2:GFP^+^ cells at the injury site at 3 dpi after nostril injury, treatment hCSF, and treatment with hCSF and double-inhibitor. *p*-values are based on one-way ANOVA (*p*-value = 1.042 × 10^−6^) with post-hoc Tukey Test (all pairs). Data are shown as mean ± SEM; each data point represents one animal. All images are full z-projections of confocal stack. Scale bars in (**C**) = 100 μm; scale bars in (**D**,**F**,**H**,**H′**,**I**, **I′**,**L**) = 20 μm. Abbreviations: dpi: days post-injury; hCSF: human cerebrospinal fluid; Inh: inhibitors; N3d; nostril 3 dpi. Symbol description: black triangle: skull injury; black circle: nostril injury; black rectangle: small skull injury; red circle: human CSF administration; blue circle: heat-inactivated hCSF treatment; light blue circle: double inhibitors, NBI 74330 and CU CPT22.

**Figure 7 cells-11-00520-f007:**
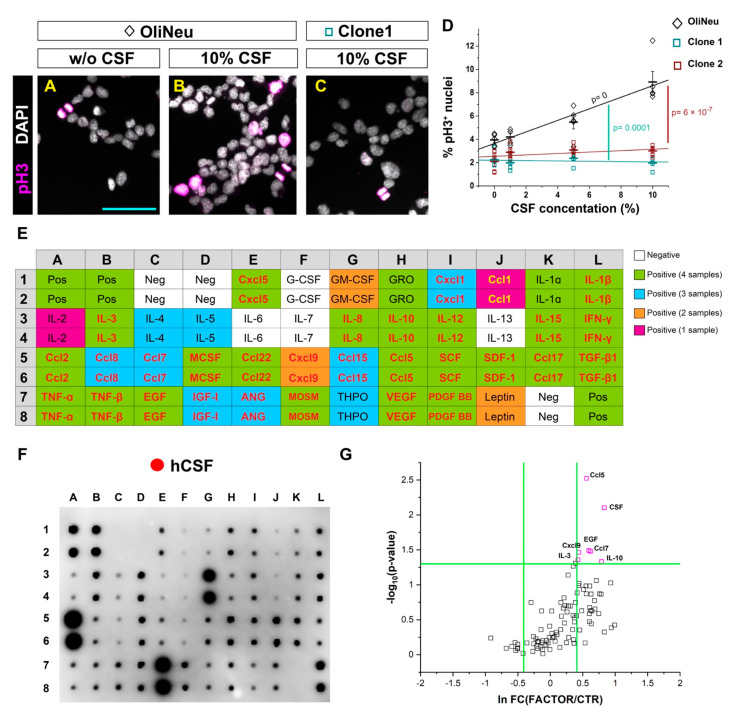
In vitro screening to identify potential candidates from the human cerebrospinal fluid inducing OPC proliferation. (**A**–**C**) Micrographs illustrating the proportion of proliferating (pH3 positive) cells in a control wildtype (WT) (**A**,**B**) and Tlr2 and Cxcr3-deficient (**C**) OliNeu oligodendrocyte progenitor cell line in basal conditions (**A**) and in response to the CSF treatment (**B**,**C**). (**D**) Dot-plot depicting the proportion of proliferating WT and Tlr2 and Cxcr3-deficient Oli-Neu cells after CSF treatment. The line indicates the corresponding linear data-fit. Data are shown as mean ± SEM; each data point represents one independent experiment. Adjusted *p*-values assess the quality of the linear fit for the WT clone (black) and difference in the slopes of the linear fits (color-coded) using the linear regression model. (**E**) Table showing the map of the array in (**F**). Color-code illustrates the presence of each cytokine in CSF samples (White: negative in all samples; Green: positive in all samples; Blue: positive in 3 out 4 samples; Orange: positive in 2 out of 4 samples; Magenta: Positive in 1 out 4 samples). Cytokines names colored in red or yellow were selected for the screening in (**G**). (**F**) Representative image of a cytokine antibody array depicting the cytokine composition of a healthy donor-derived CSF. (**G**) Dot plot depicting proliferation of Oli-Neu cells after treatment with different cytokines and CSF. Scale bars in (**A**–**C**) = 50μm Abbreviations: hCSF: human cerebrospinal fluid. Symbol description: Black diamond: control OliNeu cells; green square: Tlr2 and Cxcr3-deficient clone 1; red square Tlr2 and Cxcr3-deficient clone 2; red circle: human CSF administration.

## Data Availability

All data are available upon request.

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
