# Peer review of "Innate Immune Pathways Promote Oligodendrocyte Progenitor Cell Recruitment to the Injury Site in Adult Zebrafish Brain"

_cells, 2022, doi:10.3390/cells11030520_

Round 1
Reviewer 1 Report
In this paper, Sanchez-Gonzalez et al. investigated the mechanisms which control oligodendrocyte progenitor cells recruitment to the injury site in zebrafish. This is a multi-step study performed in a well-established model of brain injury in adult zebrafish. I have no concerns about the design and methodology. The results are sound, novel, and elegantly shown, and the only concern comes from the huge content of the manuscript.
Specific concern:
It is not explained why in the experiment designed for the identification of potential ligands activating innate immunity pathways in the CSF, the Authors used OPC cell line (OliNeu) instead of murine OPC primary cell culture which is a standard model?
Author Response
We thank our reviewer for the positive feed-back on our manuscript. In order to address the specific concern that our reviewer had we added the following statement in Material and Method section 2.27
"2.27. Screen for Cxcr3 and Tlr2 Ligands from the CSF
As screen requires many cells, we decided to conduct it in the oligodendrocyte progenitor cell line (OniNeu) that also allows us for genetic inactivation of Cxcr3 and Tlr2 as described in the section 2.26. Both....."
Reviewer 2 Report
Sanchez-Gonzalez et al present in their manuscript describe a new aspects of glial reaction in brain injury in vertebrates. According to the data presented, oligodendrocyte progenitors (OPC) can reside in the injured area short or long term based on the mode of injury, contribute to reactive gliosis and regulate neurogenic outcome. They also identified that this reaction could be independent of immuen system activity, monocytes or macrophages. They identified signaling molecules and receptors related to the phenotypes, they functionally characterized the involvement of Tlr and Cxcr pathways in this process.
The way the manuscript is written is quite informative, experiments are well designed, descriptions are comprehensive, analytical tools are appropriate. Dr. Ninkovic is a well-established and among the pioneers of the field. The content of the manuscript is quite well matching the quality level of his work.
The manuscript is acceptable in its current form. I only have a small suggestion, and if can be addressed would be informative for the reader. When human CSF is injected, can the cross-species reactivity due to misfolded protein or xenobiotic response be excluded in the study the authors performed? Heat inactivated CSF reduces the response significantly, and this is a good validation of the specific response of the CSF contents, but a couple of sentences that discuss on this point could be useful.
I would like to congratulate the authors for this comprehensive and elaborate work, which sheds a new insight into the glial biology and neural regeneration in vertebrate brains.
Author Response
We thank this reviewer for appreciation of our work. Indeed, the xenobiotic reaction was an important control that we did and presented in the current form of the manuscript. We performed two controls. First, we injected the heat-inactivated CSF and second, we injected human plasma. None of them induced the reaction of OPCs comparable to the native CSF. Due to the size of the manuscript, we did not explain these controls in details, but we have now included in the revised manuscript the following statement (lines 981-989): As we injected the human CSF, the observed reaction could be a result of xenobiotic response. Therefore, we heat-inactivated the human CSF and probed its capacity to induce the reaction of OPCs in the nostril injury. Importantly, the dramatic CSF effect was not observed upon administration of heat inactivated human CSF (Figure S8). Moreover, the administration of the human plasma containing many of the CSF components, failed to induce the response (Figure S8), indicating that prolonged OPC reactivity was not due to xenobiotic inflammation or misfolded proteins in the CSF". We that this now addresses the point raised by our reviewer.